# *Igh* and *Igk* loci use different folding principles for V gene recombination due to distinct chromosomal architectures of pro-B and pre-B cells

Louisa Hill [1], Gordana Wutz [1,3], Markus Jaritz [1,3], Hiromi Tagoh [1], Lesly Calderón [1], Jan-Michael Peters [1], Anton Goloborodko [2] & Meinrad Busslinger [1]✉

Extended loop extrusion across the immunoglobulin heavy-chain (*Igh*) locus facilitates $V_H$-$DJ_H$ recombination following downregulation of the cohesin-release factor Wapl by Pax5, resulting in global changes in the chromosomal architecture of pro-B cells. Here, we demonstrate that chromatin looping and $V_K$-$J_K$ recombination at the *Igk* locus were insensitive to Wapl upregulation in pre-B cells. Notably, the Wapl protein was expressed at a 2.2-fold higher level in pre-B cells compared with pro-B cells, which resulted in a distinct chromosomal architecture with normal loop sizes in pre-B cells. High-resolution chromosomal contact analysis of the *Igk* locus identified multiple internal loops, which likely juxtapose $V_K$ and $J_K$ elements to facilitate $V_K$-$J_K$ recombination. The higher Wapl expression in Igμ-transgenic pre-B cells prevented extended loop extrusion at the *Igh* locus, leading to recombination of only the 6 most 3' proximal $V_H$ genes and likely to allelic exclusion of all other $V_H$ genes in pre-B cells. These results suggest that pro-B and pre-B cells with their distinct chromosomal architectures use different chromatin folding principles for V gene recombination, thereby enabling allelic exclusion at the *Igh* locus, when the *Igk* locus is recombined.

V(D)J recombination at the antigen receptor loci depends on the three-dimensional chromosomal architecture, which is organized in multiple layers as revealed by high-resolution genome-wide chromosome conformation capture (Hi-C) experiments[1–3]. At the megabase scale, the chromosomes consist of compartments that reflect the segregation of transcriptionally active (type A) and inactive (type B) chromatin[4,5]. At the next level, the topologically associated domains (TADs) with a median size below 1 megabase constitute contiguous regions with a high frequency of intradomain DNA interactions that are thought to mediate the communication between promoters and enhancers[2,6,7]. The generation of TADs and their chromatin loops depends on the ring-shaped cohesin complex[8–10], which is enriched at the DNA-bound zinc finger protein CTCF in the genome[11,12]. Chromatin loops, which are generated by loop extrusion[13,14], are predominantly anchored by pairs of convergent CTCF-binding elements (CBEs) that are bound by CTCF in an orientation-dependent manner[5]. Cohesin, in association with Nipbl, functions as the loop extrusion factor[15,16], which continuously extrudes a chromatin loop until the process is halted by CTCF at the base of a loop[2]. As the cohesin-release factor Wapl determines the residence time of cohesin on chromatin[17,18], its loss

[1]Research Institute of Molecular Pathology (IMP), Vienna BioCenter (VBC), Campus-Vienna-Biocenter 1, A-1030 Vienna, Austria. [2]Institute of Molecular Biotechnology (IMBA), Austrian Academy of Sciences, Vienna BioCenter (VBC), Dr. Bohr-Gasse 3, A-1030 Vienna, Austria. [3]These authors contributed equally: Gordana Wutz, Markus Jaritz. ✉e-mail: busslinger@imp.ac.at

leads to a strong increase in loop size, demonstrating that Wapl restricts chromatin loop extension[10,19,20].

Humoral immunity to foreign pathogens depends on the generation of a diverse antigen receptor repertoire by V(D)J recombination, which assembles the variable regions of immunoglobulin (Ig) genes from variable (V), diversity (D) and joining (J) segments during B cell development[21–23]. The recombination of *Ig* genes is sequentially regulated within the B cell lineage, as the Ig heavy-chain (*Igh*) locus undergoes rearrangements in early B cell development prior to the light-chain genes (*Igk*, *Igl*) in pre-B cells[21,22]. Moreover, $D_H$-$J_H$ rearrangements at the *Igh* locus are initiated in lymphoid progenitors, followed by $V_H$-$DJ_H$ recombination in committed pro-B cells[21,22]. The mouse *Igh* locus spans 2.8 Mb and is composed of a 0.26-Mb long 3' proximal region (containing the $D_H$, $J_H$ and $C_H$ gene segments) and of a distal 2.44-Mb long $V_H$ gene cluster[24,25]. Given the large size of the $V_H$ gene cluster, contraction of the entire *Igh* locus is required to juxtapose distantly located $V_H$ genes next to the 3' proximal $DJ_H$-rearranged gene segment, which facilitates $V_H$-$DJ_H$ recombination in committed pro-B cells[26–31].

The transcription factor Pax5, which is essential for *Igh* locus contraction[27,30], was recently shown to promote prolonged chromatin loop extrusion across the entire *Igh* locus by downregulating the expression of the cohesin-release factor Wapl in pro-B cells[20]. By binding to the *Wapl* promoter, Pax5 recruits the Polycomb repressive complex 2 (PRC2), which leads to fourfold repression of *Wapl* and, consequently, to an increased residence time of cohesin on chromatin. This, in turn, causes global changes in the chromosomal architecture, as the number and length of chromatin loops are significantly increased, while the compartments are weakened in pro-B cells[20]. Prolonged loop extrusion and $V_H$ gene recombination across the entire *Igh* locus strictly depend on the following two features of the *Igh* locus. First, all CBEs in the $V_H$ gene cluster are present in the same forward direction[25] and are thus in convergent orientation to the reverse CBEs at the 3' end of the *Igh* locus, thereby facilitating loop formation[3,20]. Consequently, the inversion of CBEs in the $V_H$ gene cluster was shown to interfere with loop formation and $V_H$ gene rearrangements[20,32]. Second, all $V_H$ genes have the same forward orientation, which facilitates convergent alignment of the recognition signal sequences (RSS) of $V_H$ genes and the $DJ_H$-rearranged segment by loop extrusion in the 3' proximal RAG$^+$ recombination center[23,33] prior to RAG-mediated deletional $V_H$-$DJ_H$ recombination. As a consequence, the inversion of $V_H$ genes was shown to interfere with $V_H$-$DJ_H$ recombination in pro-B cells[20,32].

Upon successful *Igh* rearrangement in pro-B cells and subsequent transition to the pre-B cell stage, the *Igk* light-chain locus undergoes $V_K$-$J_K$ recombination at a high frequency in small pre-B cells[21], although *Igk* rearrangements can be detected at a low level already in pro-B cells[34]. The *Igk* locus, with a size of 3.2 Mb, is even larger than the *Igh* locus and contains 92 functional $V_K$ genes, four functional $J_K$ elements, and one $C_K$ region[25] (Supplementary Fig. 1a). Contraction of the *Igk* locus takes place in small pre-B cells[28] and was later shown to be initiated already in pro-B cells[35]. Importantly, the *Igk* locus differs in two key aspects from the *Igh* locus. First, most CBEs, which are largely located in the central region of the $V_K$ gene cluster, are present in the same reverse orientation as the two CBEs of the Cer element located upstream of the $J_K$ elements at the 3' end of the *Igk* locus[25,36] (Supplementary Fig. 1a), which may not favor loop formation by loop extrusion across the *Igk* locus. Second, 59 of the functional $V_K$ genes, which are mainly located in the central $V_K$ gene region, are also present in reverse orientation[25] (Supplementary Fig. 1a), which leads to inversional $V_K$-$J_K$ recombination[23]. Based on the known function of loop extrusion in controlling *Igh* recombination[20,32], the reverse orientation of both the CBEs and $V_K$ genes in the *Igk* locus appears to be incompatible with a role of extended loop extrusion in aligning the convergent RSS

sequences of all $V_K$ and $J_K$ elements prior to RAG-mediated recombination.

Here, we studied the role of Wapl expression in controlling $V_K$-$J_K$ recombination in small pre-B cells. VDJ-seq analysis revealed that the $V_K$ genes across the *Igk* locus rearranged at a similar frequency in both *Wapl*$^{\Delta P1,2/\Delta P1,2}$ and control *Wapl*$^{+/+}$ pre-B cells, although *Wapl*$^{\Delta P1,2/\Delta P1,2}$ pre-B cells exhibited a 3-fold higher expression of Wapl due to deletion of the Pax5-binding site P1 in the *Wapl* promoter[20]. Likewise, the chromosomal architecture was also similar in *Wapl*$^{\Delta P1,2/\Delta P1,2}$ and *Wapl*$^{+/+}$ pre-B cells, indicating that both $V_K$-$J_K$ recombination and the chromosomal architecture in pre-B cells are largely insensitive to Wapl expression changes. In contrast to the equally low *Wapl* mRNA levels in pro-B and pre-B cells[20], we unexpectedly discovered that the Wapl protein was expressed at a 2.2-fold higher level in pre-B cells compared with pro-B cells, which resulted in a distinct chromosomal architecture with smaller loop sizes in pre-B cells compared with the pro-B cell architecture characterized by extended loops. High-resolution mapping of interactions in pre-B cells revealed that the contraction of the *Igk* locus[28] is caused by the formation of multiple internal loops, which likely juxtaposes $V_K$ and $J_K$ elements to facilitate $V_K$-$J_K$ recombination. Notably, the higher Wapl expression in Igμ-transgenic pre-B cells interfered with extended loop extrusion at the *Igh* locus, leading to recombination of only the 6 most 3' proximal $V_H$ genes and likely to the allelic exclusion of all other $V_H$ genes in pre-B cells. Together, our data demonstrate that the *Igh* and *Igk* loci use distinct folding principles for V gene recombination due to the different chromosomal architectures of pro-B and pre-B cells.

## Results

### Efficient $V_K$ gene recombination across the *Igk* locus in pre-B cells with high Wapl expression

The starting point for our study was the previous finding that *Wapl* mRNA is fourfold downregulated in both pro-B and pre-B cells compared with uncommitted progenitors and mature B cells[20]. This observation raised the question of whether *Wapl* repression is also essential for $V_K$ gene recombination at the *Igk* locus as recently shown for the *Igh* locus[20]. To address this issue, we took advantage of the *Wapl*$^{\Delta P1,2/\Delta P1,2}$ mouse, which lacks the functional Pax5-binding site P1 in the *Wapl* promoter as well as a hypersensitive region downstream of *Wapl*, containing a second Pax5-binding site (P2) that is not required for *Wapl* expression in early B cells[20]. As *Wapl* mRNA expression is known to be increased fourfold in *Wapl*$^{\Delta P1,2/\Delta P1,2}$ pre-B cells compared with wild-type pre-B cells[20], we next analyzed the expression of the Wapl protein by immunoblot analysis of ex vivo sorted *Wapl*$^{\Delta P1,2/\Delta P1,2}$ and control *Wapl*$^{+/+}$ pre-B cells (CD19$^+$B220$^+$IgM$^-$IgD$^-$Kit$^-$CD25$^+$) from the bone marrow (Supplementary Fig. 2a). Consistent with the observed *Wapl* mRNA increase, the Wapl protein was threefold more highly expressed in *Wapl*$^{\Delta P1,2/\Delta P1,2}$ pre-B cells relative to *Wapl*$^{+/+}$ pre-B cells (Fig. 1a). As elevated Wapl expression can lower the residence time of cohesin on chromatin[20], the observed Wapl increase may lead to impaired loop extrusion and thus potential defects in $V_K$-$J_K$ rearrangement in *Wapl*$^{\Delta P1,2/\Delta P1,2}$ pre-B cells. Unexpectedly however, systematic analysis of $V_K$-$J_K$ recombination by the VDJ-seq method[37] revealed that individual $V_K$ genes along the entire *Igk* locus rearranged at a largely similar frequency in *Wapl*$^{\Delta P1,2/\Delta P1,2}$ and *Wapl*$^{+/+}$ pre-B cells (Fig. 1b and Supplementary Data 1a), regardless of their forward or reverse orientation (Supplementary Fig. 1b). Even the most distal $V_K$ genes ($V_K$2-137, $V_K$1-135) at the 5' end of the *Igk* locus recombined at similar frequencies in both pre-B cell types. Differences in recombination frequency were primarily observed for the most 3' proximal $V_K$ genes ($V_K$3-2 to $V_K$3-12) and some central $V_K$ genes ($V_K$4-79 to $V_K$10-96) that rearranged more efficiently in *Wapl*$^{\Delta P1,2/\Delta P1,2}$ pre-B cell, while certain distal $V_K$ genes ($V_K$9-120 to $V_K$11-133) rearranged better in *Wapl*$^{+/+}$ pre-B cells (Fig. 1b and Supplementary Fig. 1c).

The majority of primary $V_K$-$J_K$ rearrangements is known to involve the 5' most $J_K1$ element, while the downstream $J_K2$, $J_K4$ and $J_K5$ elements are mainly used for secondary $V_K$-$J_K$ recombination occurring during receptor editing[38,39]. The overall frequencies of $V_K$ rearrangements involving the 4 functional $J_K$ elements were equivalent between $Wapl^{\Delta P1,2/\Delta P1,2}$ and $Wapl^{+/+}$ pre-B cells (Supplementary Fig. 1d) and were comparable to published data[40,41]. Moreover, the recombination frequencies of individual $V_K$ genes involving the $J_K1$ or $J_K5$ element were also largely similar in $Wapl^{\Delta P1,2/\Delta P1,2}$ and $Wapl^{+/+}$ pre-B cells (Fig. 1c and Supplementary Fig. 1e), further indicating that a threefold increase of

Wapl expression had only a minimal effect on $V_K$-$J_K$ recombination across the *Igk* locus.

We next used Hi-C sequencing[5] to study the long-range interactions at the *Igk* locus in ex vivo sorted pre-B cells from the bone marrow of $Wapl^{\Delta P1,2/\Delta P1,2}$ and $Wapl^{+/+}$ mice (Supplementary Fig. 2a). The Hi-C contact map of the *Igk* locus in $Wapl^{+/+}$ pre-B cells revealed that the *Igk* 3' region, containing the $J_K$ elements and *Igk* enhancers, interacted with sequences across the entire *Igk* locus and that the large *Igk* TAD consisted of sub-TADs (Fig. 1d), as previously reported in ref. 42. Surprisingly, the Hi-C contact maps of $Wapl^{\Delta P1,2/\Delta P1,2}$ and $Wapl^{+/+}$ pre-B cells

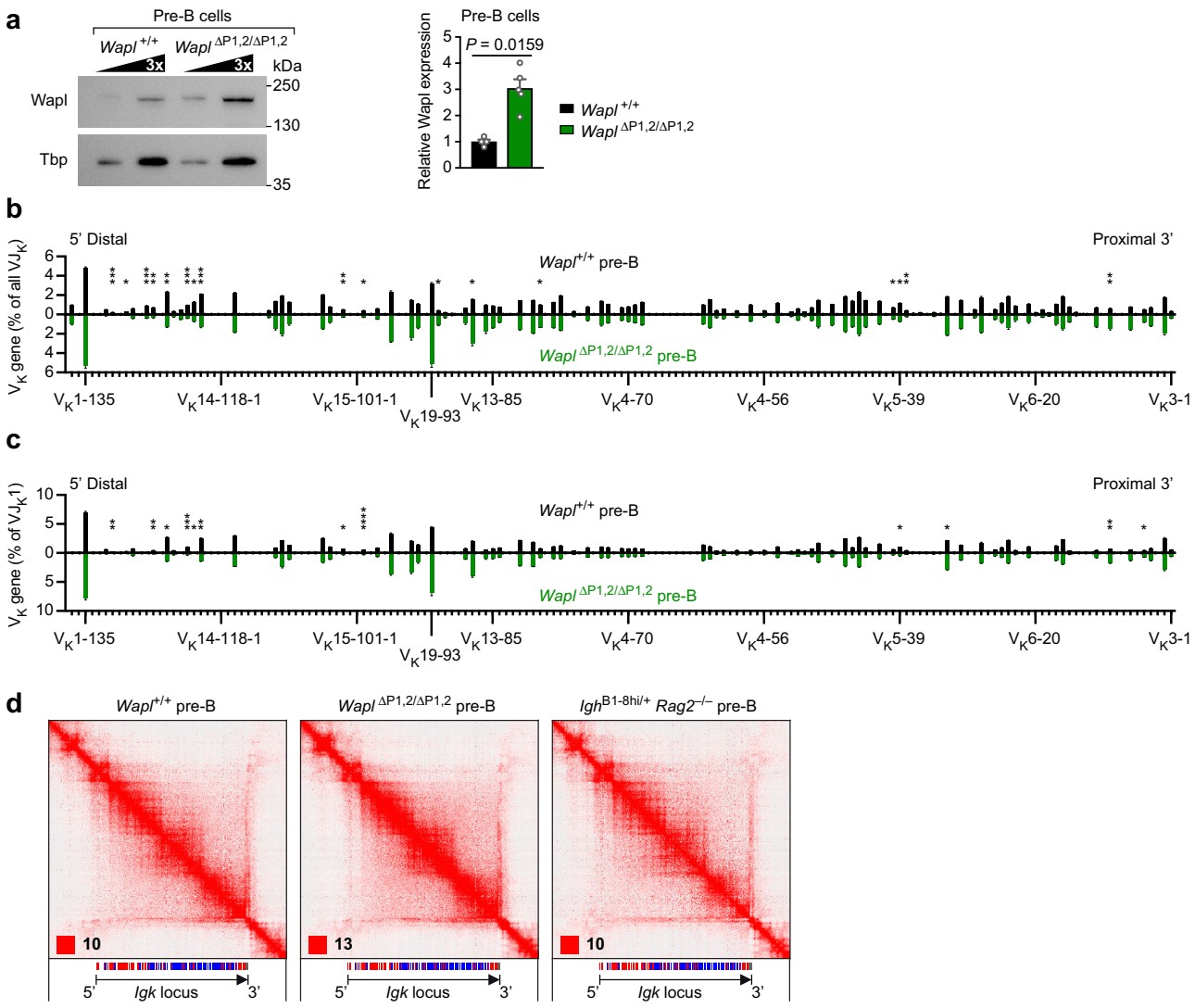

**Fig. 1 | Efficient $V_K$-$J_K$ rearrangements and long-range interactions across the *Igk* locus upon high Wapl expression in $Wapl^{\Delta P1,2/\Delta P1,2}$ pre-B cells. a** Wapl expression in ex vivo sorted pre-B cells from the bone marrow of $Wapl^{+/+}$ (black) and $Wapl^{\Delta P1,2/\Delta P1,2}$ (green) mice, as determined by immunoblot analysis of threefold serially diluted whole-cell extracts with antibodies detecting Wapl or the TATA-binding protein (Tbp; loading control). Marker proteins of the indicated size (in kilodaltons, kDa) are shown to the right. The quantification of four independent immunoblot experiments is indicated to the right. Statistical data are shown as mean values with SEM and were analyzed with the Student's *t*-test (unpaired and two-tailed). **b, c** $V_K$ gene recombination analysis of ex vivo sorted $Wapl^{+/+}$ and $Wapl^{\Delta P1,2/\Delta P1,2}$ pre-B cells, as determined by VDJ-seq experiments[37]. The VDJ-seq data obtained with $Wapl^{+/+}$ (black) and $Wapl^{\Delta P1,2/\Delta P1,2}$ (green) pre-B cells are shown in the upper and lower part, respectively. The recombination frequency of each $V_K$ gene is indicated as a percentage of all $V_K$-$J_K$ (**b**) or only the $V_K$-$J_K1$ (**c**) rearrangements and is

shown as mean value with SEM based on four independent VDJ-seq experiments for each pre-B cell type. The different $V_K$ genes (horizontal axis) are aligned according to their position in the *Igk* locus[25] (Supplementary Data 1a). Statistical data were analyzed by multiple *t*-tests (unpaired and two-tailed) with Holm–Sidak correction; *$P < 0.05$, **$P < 0.01$, ***$P < 0.001$, ****$P < 0.0001$. **d** Hi-C contact matrices of the *Igk* region on chromosome 6, which were determined for ex vivo sorted $Wapl^{+/+}$, $Wapl^{\Delta P1,2/\Delta P1,2}$, and $Igh^{B1-8hi/+}$ $Rag2^{-/-}$ pre-B cells. The orientation and annotation of the *Igk* locus are shown. The intensity of each pixel represents the normalized number of contacts between a pair of loci[5]. The maximum intensity is indicated in the lower left of each panel (red square). The following resolution of the Hi-C data was calculated as described in Methods; 6.65 kb ($Wapl^{+/+}$ pre-B cells), 4.25 kb ($Wapl^{\Delta P1,2/\Delta P1,2}$ pre-B cells), and 11.8 kb ($Igh^{B1-8hi/+}$ $Rag2^{-/-}$ pre-B cells). One of two Hi-C experiments performed with pre-B cells of each genotype is shown. The VDJ-seq and Hi-C-seq data are further described in Supplementary Data 3. Source data are provided in the Source data file.

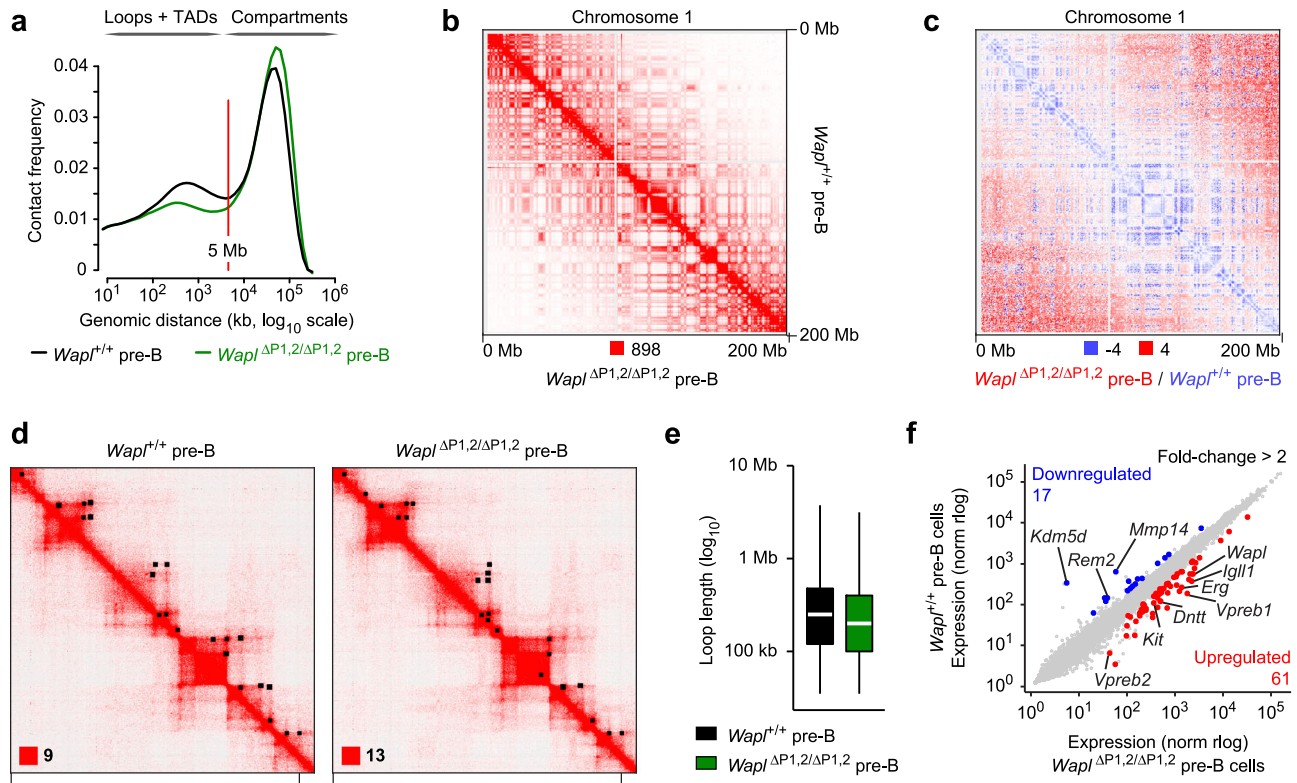

**Fig. 2 | Minor differences in the chromosomal architecture of *Wapl*[+/+] and *Wapl*[ΔP1,2/ΔP1,2] pre-B cells. a** Frequency distribution of intrachromosomal contacts as a function of the genomic distance using logarithmically increased genomic distance bins, determined by Hi-C analysis of ex vivo sorted *Wapl*[+/+] (black) and *Wapl*[ΔP1,2/ΔP1,2] (green) pre-B cells using HOMER (see Methods). **b** Hi-C contact matrices of chromosome 1, determined for ex vivo sorted *Wapl*[+/+] and *Wapl*[ΔP1,2/ΔP1,2] pre-B cells and plotted at a 500-kb bin resolution with Juicebox[74]. The intensity of each pixel represents the normalized number of contacts between a pair of loci[5]. The maximum pixel intensity is indicated below (red square). **c** Differential Hi-C contact matrix of chromosome 1 displaying the difference in pixel intensity between *Wapl*[+/+] pre-B and *Wapl*[ΔP1,2/ΔP1,2] pre-B cells as the ratio (*Wapl*[ΔP1,2/ΔP1,2])/(*Wapl*[+/+]). More interactions (blue) in the TAD range were observed for *Wapl*[+/+] pro-B cells, while more interactions (red) in the compartment range were detected for *Wapl*[ΔP1,2/ΔP1,2] pre-B cells. **d** Hi-C contact matrices of a zoomed-in region on chromosome 12

(mm9; 77,500,000–82,500,000), shown for *Wapl*[+/+] and *Wapl*[ΔP1,2/ΔP1,2] pre-B cells. Black dots indicate loop anchors identified with Juicebox. The maximum pixel intensity is indicated in the lower left of the panel (red square). **e** Distribution of the loop length (in kb). White lines indicate the median and boxes represent the middle 50% of the data. Whiskers denote all values of the 1.5× interquartile range. The median loop length is 250 kb in *Wapl*[+/+] pre-B cells (black) and 200 kb in *Wapl*[ΔP1,2/ΔP1,2] pre-B cells (green). **f** Scatterplot of gene expression differences between *Wapl*[ΔP1,2/ΔP1,2] and *Wapl*[+/+] pre-B cells isolated from the bone marrow of the respective mice. Genes upregulated (red) or downregulated (blue) in *Wapl*[ΔP1,2/ΔP1,2] pre-B cell compared with *Wapl*[+/+] pre-B cells were analyzed with DESeq2 and defined by an expression difference of >2-fold, an adjusted *P* value of <0.05 and a TPM value of >5 in at least one of the two pre-B cell types (Supplementary Data 2). Two RNA-seq experiments were performed with ex vivo sorted pre-B cells of each genotype.

were comparable, which was confirmed by the quantification of the Hi-C interaction frequencies at the *Igk* locus (Supplementary Fig. 3a–c). Hence, the *Igk* locus has a similar TAD structure in both pre-B cell types (Fig. 1d). We next performed Hi-C analysis with *Igh*[B1-8hi/+] *Rag2*[−/−] pre-B cells, which were generated in the absence of V(D)J recombination due to RAG2 loss by skipping the pro-B cell stage through the expression of the functionally pre-rearranged *Igh* gene B1-8[hi] (ref. 43). Notably, the Hi-C contact map of *Igh*[B1-8hi/+] *Rag2*[−/−] pre-B cells was similar to those of *Wapl*[+/+] and *Wapl*[ΔP1,2/ΔP1,2] pre-B cells (Fig. 1d and Supplementary Fig. 3a–c), which indicated that the Hi-C stripe extending from the 3′ end across the entire *Igk* locus is caused by long-range interactions rather than by V_K-J_K rearrangements. We conclude, therefore, that increased Wapl expression in *Wapl*[ΔP1,2/ΔP1,2] pre-B cells had a minimal effect on long-range interactions and V_K gene rearrangements at the *Igk* locus, which is in marked contrast to the exquisite sensitivity of *Igh* V_H-DJ_H recombination to elevated Wapl levels in *Wapl*[ΔP1,2/ΔP1,2] pro-B cells[20].

**Increased Wapl expression minimally affects the chromosomal architecture of pre-B cells**

We next analyzed the chromosomal architecture of the entire genome in *Wapl*[+/+] and *Wapl*[ΔP1,2/ΔP1,2] pre-B cells by interrogating the Hi-C

data. Analysis of all identified sequence contacts within the genome revealed that the frequencies of intrachromosomal contacts up to a distance of 5 Mb, which largely generate chromatin loops within TADs[5], were moderately decreased in *Wapl*[ΔP1,2/ΔP1,2] pre-B cells compared with *Wapl*[+/+] pre-B cells (Fig. 2a). Moreover, the frequencies of contacts over very large distances (>10 Mb), which largely reflect chromosomal compartments[4], were modestly increased in *Wapl*[ΔP1,2/ΔP1,2] pre-B cells relative to *Wapl*[+/+] pre-B cells (Fig. 2a). Consistent with this finding, visual inspection of the Hi-C contact map of chromosome 1 revealed a well-defined checkerboard pattern for both *Wapl*[ΔP1,2/ΔP1,2] and *Wapl*[+/+] pre-B cells (Fig. 2b). Modest differences are best highlighted by a differential contact map, which displays the difference in contact frequencies between *Wapl*[+/+] and *Wapl*[ΔP1,2/ΔP1,2] pre-B cells (Fig. 2c). This analysis confirmed a relative enrichment of short-range interactions in the TADs of *Wapl*[+/+] pre-B cells, while long-range interactions in the compartment range were increased in *Wapl*[ΔP1,2/ΔP1,2] pre-B cells. However, these minimal differences had no apparent effect on TAD structures and chromatin looping in *Wapl*[ΔP1,2/ΔP1,2] pre-B cells, as shown for a zoomed-in region on chromosome 12 (Fig. 2d). This finding is consistent with the observed minor decrease of the median loop length from 250 kb in *Wapl*[+/+] pre-B cells to 200 kb in *Wapl*[ΔP1,2/ΔP1,2] pre-B cells (Fig. 2e and Supplementary Fig. 4a) and with the minimal difference in loop

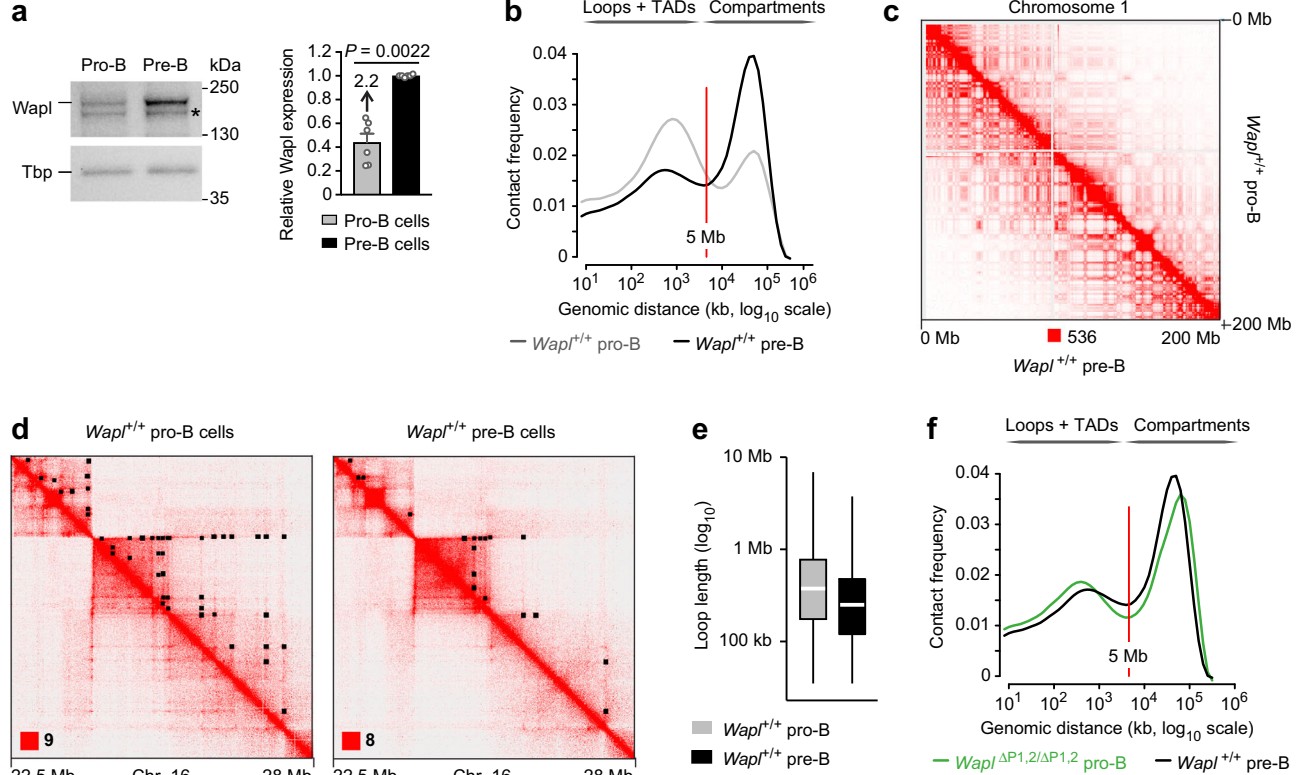

**Fig. 3 | Wild-type pro-B and pre-B cells strongly differ in their chromosomal architecture. a** Wapl protein expression in ex vivo sorted *Rag2*[−/−] pro-B (gray) and wild-type pre-B (black) cells from the bone marrow, as determined by immunoblot analysis of whole-cell extracts with antibodies detecting Wapl or Tbp (loading control). Marker proteins of the indicated size (in kilodaltons, kDa) are shown to the right. A non-specific band is indicated by an asterisk. One of 5 independent experiments is shown, and the quantification of all immunoblot experiments is indicated to the right. Statistical data are shown as mean values with SEM and were analyzed with the Student's *t*-test (unpaired and two-tailed). **b** Comparison of the frequency distribution of intrachromosomal contacts between pro-B and pre-B cells of *Wapl*[+/+] mice. The frequency distribution of intrachromosomal contacts was determined as a function of the genomic distance using logarithmically increased genomic distance bins, based on the Hi-C data of *Wapl*[+/+] pre-B cells (black, this

study) and published Hi-C data of short-term cultured *Wapl*[+/+] pro-B cells (gray)[20]. **c** Hi-C contact matrices of chromosome 1, determined for *Wapl*[+/+] pro-B and *Wapl*[+/+] pre-B cells, were plotted at a 500-kb bin resolution with Juicebox[74]. The maximum pixel intensity is indicated below (red square). **d** Hi-C contact matrices of a zoomed-in region on chromosome 16 (mm9; 22,500,000–28,000,000), shown for *Wapl*[+/+] pro-B and pre-B cells. Black dots indicate loop anchors identified with Juicebox[74]. **e** Distribution of the loop length (in kb) in *Wapl*[+/+] pro-B (gray) and pre-B (black) cells. White lines indicate the median and boxes represent the middle 50% of the data. Whiskers denote all values of the 1.5× interquartile range. The median loop length is 375 kb in *Wapl*[+/+] pro-B cells and 250 kb in *Wapl*[+/+] pre-B cells. **f** Comparison of the frequency distribution of intrachromosomal contacts between *Wapl*[+/+] pre-B cells (black) and *Wapl*[ΔP1,2/ΔP1,2] pro-B cells (green)[20], as described in (**b**). Source data are provided in the Source Data file.

numbers determined for the two pre-B cell types (Supplementary Fig. 4b). Together, these data indicate that a threefold increase of Wapl protein expression had a relatively minor effect on the chromosomal architecture of pre-B cells in marked contrast to the strong effects elicited by a similar increase of Wapl protein expression in pro-B cells[20].

To explore whether the observed architectural changes affect gene expression, we analyzed ex vivo sorted *Wapl*[ΔP1,2/ΔP1,2] and *Wapl*[+/+] pre-B cells by RNA-sequencing, which identified 61 upregulated and 17 downregulated genes with an expression difference of >2-fold in *Wapl*[ΔP1,2/ΔP1,2] pre-B cells relative to *Wapl*[+/+] pre-B cells (Fig. 2f and Supplementary Data 2). The differentially expressed genes code for proteins of distinct functional classes, including several surface proteins, signal transducers, and metabolic enzymes (Supplementary Fig. 4c and Supplementary Data 2). Notably, the genes encoding the surrogate light chains VpreB1, VpreB2, and Igλ (*Igll1*), the surface receptor Kit, and the terminal deoxynucleotidyl transferase (*Dntt*) were expressed at a higher level in *Wapl*[ΔP1,2/ΔP1,2] pre-B cells compared with *Wapl*[+/+] pre-B cells (Fig. 2f), where these genes are normally downregulated in response to pre-B cell receptor signaling[44]. The same expression analysis of *Wapl*[ΔP1,2/ΔP1,2] and *Wapl*[+/+] pro-B cells previously identified a higher number of differentially expressed genes, as 161 genes were upregulated and 159 genes were downregulated upon

increased Wapl expression in pro-B cells[20]. There was, however, only a small overlap between the differentially expressed genes in pro-B and pre-B cells, with 16 upregulated and eight downregulated genes being present in both datasets (Supplementary Fig. 4d). These data, therefore, indicated that increased Wapl protein expression had a smaller effect on both differential gene expression and genomic architecture in pre-B cells relative to pro-B cells.

## Wild-type pro-B and pre-B cells strongly differ in their chromosomal architecture

We next investigated the Wapl protein levels in ex vivo sorted pro-B and pre-B cells of *Wapl*[+/+] mice by immunoblot analysis, which revealed that Wapl was expressed at a 2.2-fold higher level in pre-B cells compared with pro-B cells (Fig. 3a). This result was unexpected, as the *Wapl* mRNA is expressed at the same low level in pro-B and pre-B cells and is increased 4-fold only in immature and mature B cells[20] (Supplementary Fig. 4e). The discrepancy between *Wapl* mRNA and Wapl protein expression in pre-B cells may indicate that the *Wapl* mRNA is either more efficiently translated in pre-B cells or that the Wapl protein is stabilized by a yet unknown posttranslational mechanism in pre-B cells. As predicted by the 2.2-fold difference in Wapl protein expression between wild-type pro-B and pre-B cells, the chromosomal architecture

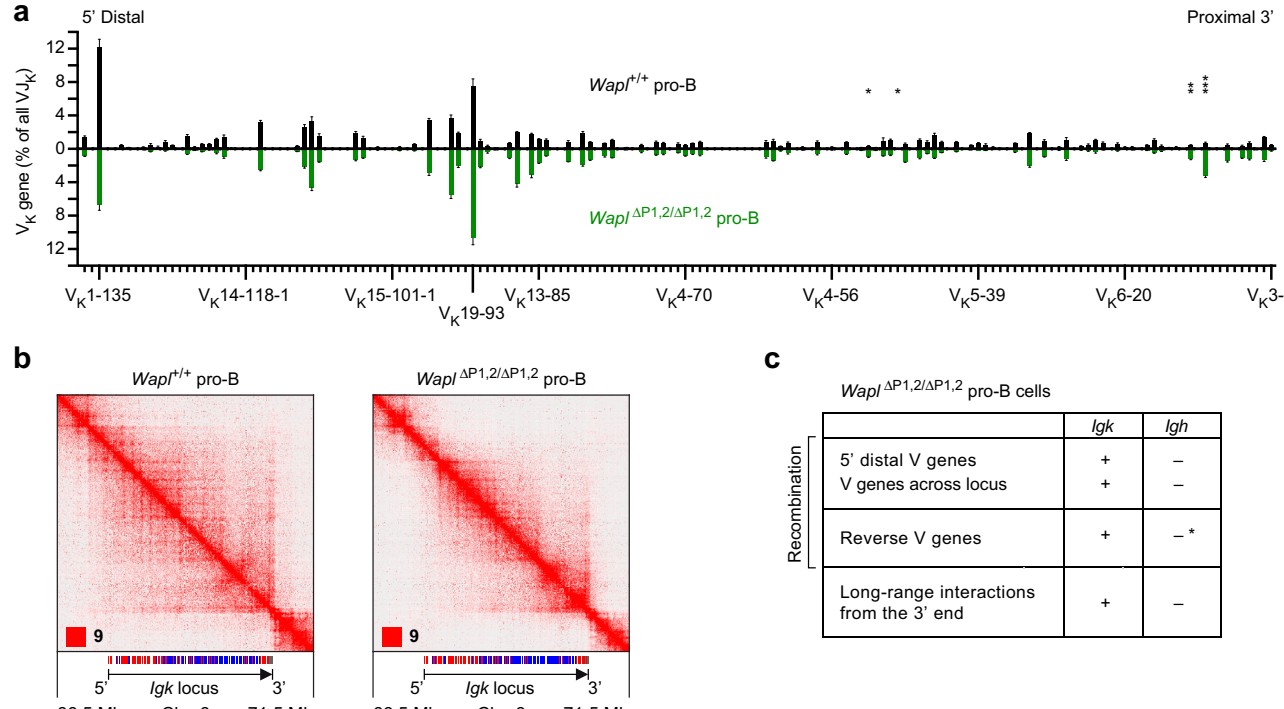

**Fig. 4 | V$_K$-J$_K$ recombination at the *Igk* locus in Wapl$^{high}$ and Wapl$^{low}$ pro-B cells.**
**a** V$_K$ gene recombination analysis of ex vivo sorted *Wapl*$^{+/+}$ (Wapl$^{low}$; black) and *Wapl*$^{\Delta P1,2/\Delta P1,2}$ (Wapl$^{high}$; green) pro-B cells, as determined by VDJ-seq experiments. The recombination frequency of each V$_K$ gene is indicated as a percentage of all V$_K$-J$_K$ rearrangement events and is shown as a mean value with SEM based on six independent VDJ-seq experiments for each pro-B cell type. The different V$_K$ genes (horizontal axis) are aligned according to their position in the *Igk* locus[25] (Supplementary Data 1a). Statistical data were analyzed by multiple *t*-tests (unpaired and two-tailed) with Holm–Sidak correction; *$P < 0.05$, **$P < 0.01$, ***$P < 0.001$. **b** Hi-C contact matrices of the *Igk* region on chromosome 6 based on published Hi-C data

of short-term cultured *Wapl*$^{+/+}$ and *Wapl*$^{\Delta P1,2/\Delta P1,2}$ pro-B cells[20]. The orientation and annotation of the *Igk* locus are shown. The resolution of the Hi-C data was 6.7 and 7.25 kb for the *Wapl*$^{+/+}$ and *Wapl*$^{\Delta P1,2/\Delta P1,2}$ pro-B cells, respectively. **c** Differences in V gene recombination and long-range interactions between the *Igh* and *Igk* locus in *Wapl*$^{\Delta P1,2/\Delta P1,2}$ pro-B cells, based on the data shown in Fig. 4 and published data[20]. The loss of recombination of V$_H$ genes upon their inversion (indicated by an asterisk) in the *Igh* locus was analyzed in *Igh*$^{V8-8-inv/V8-8-inv}$ and *Igh*$^{\Delta 890/\Delta 890}$ pro-B cells[20] and upon inversion of the entire V$_H$ gene cluster in *v-Abl* immortalized pro-B cells[32]. Source data are provided in the Source Data file.

differed considerably between the two cell types, as indicated by the significantly lower frequencies of intrachromosomal contacts in the TAD range (<5 Mb) and the strongly increased frequencies of long-distance contacts in the compartment range (>10 Mb) in *Wapl*$^{+/+}$ pre-B cells compared with *Wapl*$^{+/+}$ pro-B cells (Fig. 3b). In this context, it is important to note that lower Wapl expression is known to cause a breakdown of compartmentalization, as longer loops interfere with the compartment structure[10,19]. Consistent with this finding, visual inspection of the Hi-C contact map of chromosome 1 revealed a well-defined checkerboard pattern (Fig. 3c), caused by increased compartmentalization (Supplementary Fig. 4f), in Wapl$^{high}$ *Wapl*$^{+/+}$ pre-B cells relative to Wapl$^{low}$ *Wapl*$^{+/+}$ pro-B cells. Moreover, a zoomed-in region on chromosome 16 revealed that the long extension of loops in *Wapl*$^{+/+}$ pro-B cells was not detected in *Wapl*$^{+/+}$ pre-B cells (Fig. 3d), which is consistent with the observed decrease of the median loop length from 375 kb in *Wapl*$^{+/+}$ pro-B cells[20] to 250 kb in *Wapl*$^{+/+}$ pre-B cells (Fig. 3e) and a 1.8-fold decrease in loop numbers from *Wapl*$^{+/+}$ pro-B cells to *Wapl*$^{+/+}$ pre-B cells (Supplementary Fig. 4g). Importantly, these findings revealed an exquisite sensitivity of loop formation on Wapl dosage, as a 2.2-fold increase of Wapl protein expression from *Wapl*$^{+/+}$ pro-B to *Wapl*$^{+/+}$ pre-B cells resulted in drastic changes of the chromosomal architecture and loop length, while an additional 3-fold increase of Wapl protein expression from *Wapl*$^{+/+}$ pre-B cells to *Wapl*$^{\Delta P1,2/\Delta P1,2}$ pre-B cells had only a minimal effect on chromosomal architecture and loop length (Supplementary Fig. 4h).

Notably, *Wapl*$^{\Delta P1,2/\Delta P1,2}$ pro-B cells and *Wapl*$^{+/+}$ pre-B cells exhibited a similar chromosomal architecture, which was manifested by their

comparable contact frequency distributions of intrachromosomal contacts (Fig. 3f), similar patterns of A- and B-type compartmentalization (Supplementary Fig. 5a, b) and comparable loop numbers and lengths (Supplementary Fig. 5c, d). The observed similarities are consistent with a similar Wapl protein increase in *Wapl*$^{\Delta P1,2/\Delta P1,2}$ pro-B cells (threefold) and *Wapl*$^{+/+}$ pre-B cells (2.2-fold) compared with *Wapl*$^{+/+}$ pro-B cells (Fig. 3a)[20]. Together these data, therefore, demonstrate that wild-type pro-B and pre-B cells strongly differ in their chromosomal architecture, which is likely caused by the observed 2.2-fold difference in Wapl protein expression between the two cell types.

## V$_K$ gene recombination across the *Igk* locus in both Wapl$^{high}$ and Wapl$^{low}$ pro-B cells

As pro-B cells have a distinct chromosomal architecture, already undergo contraction of the *Igk* locus[35] and rearrange V$_K$ genes at a low frequency[34,41], we next investigated the V$_K$ gene recombination pattern by VDJ-seq in ex vivo sorted pro-B cells (CD19$^+$B220$^+$IgM$^-$IgD$^-$Kit$^+$CD25$^-$) from the bone marrow of *Wapl*$^{+/+}$ and *Wapl*$^{\Delta P1,2/\Delta P1,2}$ mice (Supplementary Fig. 2b). Notably, V$_K$ gene recombination was observed along the entire *Igk* locus and involved the different J$_K$ elements at a similar frequency in both pro-B cell types, regardless of low (*Wapl*$^{+/+}$) or high (*Wapl*$^{\Delta P1,2/\Delta P1,2}$) Wapl expression and independent of the V$_K$ gene orientation (Fig. 4a and Supplementary Fig. 6a–c). The V$_K$ gene recombination pattern was quite similar between *Wapl*$^{+/+}$ and *Wapl*$^{\Delta P1,2/\Delta P1,2}$ pro-B cells (Fig. 4a). However, the most 3′ proximal V$_K$ genes (V$_K$3-4 to V$_K$3-12) and central V$_K$ genes (V$_K$4-86 to V$_K$10-95) rearranged more efficiently in *Wapl*$^{\Delta P1,2/\Delta P1,2}$ pro-B cells compared with

$Wapl^{+/+}$ pro-B cells (Supplementary Fig. 6b) similar to the observed increased recombination frequency of these $V_K$ genes in $Wapl^{\Delta P1,2/\Delta P1,2}$ pre-B cells relative to $Wapl^{+/+}$ pre-B cells (Supplementary Fig. 1c). Notably, a direct comparison of the $V_K$ gene recombination pattern between $Wapl^{+/+}$ pro-B and $Wapl^{+/+}$ pre-B cells revealed that the $V_K$ genes in the 3' half of the $Igk$ locus up to the $V_K$4-79 gene rearranged better in $Wapl^{+/+}$ pre-B cells, while the more distal $V_K$ genes recombined more efficiently in $Wapl^{+/+}$ pro-B cells (Supplementary Fig. 6d), as previously described[41]. Importantly, the most distal $V_K$ genes ($V_K$2-137, $V_K$1-135), which are present in forward orientation at the 5' end of the $Igk$ locus, were still able to recombine in $Wapl^{\Delta P1,2/\Delta P1,2}$ pro-B cells (Fig. 4a), while distal $V_H$ genes (also in forward orientation) at the 5' end of the $Igh$ locus fail to recombine in the very same pro-B cells[20]. These data, therefore, suggest that the mechanism of chromatin folding at the $Igk$ and $Igh$ loci must be fundamentally different in pro-B cells.

Hi-C contact maps revealed long-range interactions from the $Igk$ 3' region across the entire $Igk$ locus in $Wapl^{+/+}$ pro-B cells (Fig. 4b) similar to $Wapl^{+/+}$ pre-B cells (Fig. 1d). The structures of the sub-TADs were, however, less well defined in $Wapl^{+/+}$ pro-B cells compared with $Wapl^{\Delta P1,2/\Delta P1,2}$ pro-B cells as the low Wapl expression in $Wapl^{+/+}$ pro-B cells resulted in a significant extension of loops within the $Igk$ TAD in these cells (Fig. 4b). Consequently, the sub-TAD structures in the $V_K$ gene cluster differed between $Wapl^{+/+}$ and $Wapl^{\Delta P1,2/\Delta P1,2}$ pro-B cells, as shown by quantification of the Hi-C interaction frequencies at the $Igk$ locus (Supplementary Fig. 3d, f). Surprisingly, the long-range interactions from the $Igk$ 3' region along the $Igk$ locus were still formed in $Wapl^{\Delta P1,2/\Delta P1,2}$ pro-B cells, although at a lower frequency compared with $Wapl^{+/+}$ pro-B cells (Fig. 4b and Supplementary Fig. 3d, e). Finally, a direct comparison of the Hi-C contact maps of the $Igk$ and $Igh$ loci in $Wapl^{\Delta P1,2/\Delta P1,2}$ pro-B cells by visual inspection highlighted the fact that long-range interactions from the 3' end were observed across the $Igk$ locus but were absent along the $Igh$ locus in these $Wapl^{high}$ pro-B cells (Supplementary Fig. 6e).

In summary, the V gene rearrangements at the $Igk$ and $Igh$ loci differ in three fundamental aspects in $Wapl^{\Delta P1,2/\Delta P1,2}$ pro-B cells. First, V genes rearrange across the entire $Igk$ locus in these $Wapl^{high}$ pro-B cells in marked contrast to the $Igh$ locus (Fig. 4a, c). Second, reverse-oriented V genes undergo rearrangements at the $Igk$ locus but fail to recombine in the context of the $Igh$ locus in pro-B cells (Fig. 4c and Supplementary Fig. 6a). Third, long-range interactions from the 3' end occur only at the $Igk$ locus but not at the $Igh$ locus in $Wapl^{\Delta P1,2/\Delta P1,2}$ pro-B cells (Fig. 4b, c). We conclude therefore that a different chromatin folding principle must operate at the $Igk$ locus to promote $V_K$ recombination as opposed to the extended loop extrusion model that explains the convergent alignment of RSS sequences of the $V_H$ genes and $DJ_H$-rearranged element prior to $V_H$-$DJ_H$ recombination at the $Igh$ locus[20].

## The $V_K$ gene region contracts in pre-B cells by folding into multiple different loops

The recently developed Micro-C method, which relies on micrococcal nuclease digestion of fixed chromatin, facilitates genome-wide analysis of the fine-scale chromatin organization at nucleosomal resolution[45,46]. We next employed Micro-C analysis to study the chromatin folding along the $Igk$ locus at high resolution in ex vivo sorted $Rag2^{-/-}$ pro-B and $Igh^{B1-8hi/+}$ $Rag2^{-/-}$ pre-B cells (Fig. 5a, b and Supplementary Fig. 7a, b). Notably, the Micro-C contact matrices at the $Igk$ locus differed significantly from each other in pro-B and pre-B cells. In pro-B cells, the $Igk$ locus consisted of two different TADs, which were present in distinct regions of less accessible compartment B that were separated by a small stretch of transcriptionally active compartment A (Fig. 5a) centered at the E88 enhancer[42]. In contrast, compartment A is present throughout the entire $Igk$ locus in pre-B cells (Fig. 5b). As the interactions from the 3' end across the $Igk$ locus appeared to differ between pro-B and pre-B cells, we quantified the interaction

frequencies along this stripe (Supplementary Fig. 7c). Notably, the interaction frequencies were specifically increased in the distal half of the $Igk$ locus in $Rag2^{-/-}$ pro-B cells compared with $Igh^{B1-8hi/+}$ $Rag2^{-/-}$ pre-B cells (Supplementary Fig. 7c), which may explain the observed preferential $V_K$ gene usage in the distal $Igk$ region in $Rag2^{+/+}$ pro-B cells relative to $Rag2^{+/+}$ pre-B cells (Supplementary Figs. 6d, 7d).

Interestingly, a high degree of substructure was observed in the 5' distal and central regions of the $V_K$ gene cluster in pre-B cells, while the density of interactions between these substructures and the stripe emanating from the 3' proximal Cer region was quite low in these cells (Fig. 5b). To map the interactions causing the observed substructures at the $Igk$ locus, we analyzed the Micro-C data with the Cross-score algorithm that quantifies the frequency of upstream and downstream long-distance contacts for each genomic site, thus measuring its ability to anchor genomic loops (Supplementary Fig. 7a, b, Methods). Peak calling on the upstream and downstream Cross-score profiles identified at least 17 peaks, which colocalized with the observed stripes in the Micro-C pattern of the $Igk$ locus (Supplementary Fig. 7b). Notably, the Cross-score peaks mapped to CBEs of matching orientation (i.e., peaks in the downstream Cross-score profile matched forward CBEs, and vice versa), suggesting that these peaks located at the interaction stripes are caused by CTCF-mediated anchoring of cohesin loops (Fig. 5c and Supplementary Fig. 7a). These data, therefore, demonstrate that the forward and reverse CBEs are responsible for the formation of multiple different loops along the $V_K$ gene region. Interestingly, a deep gap was observed in both Cross-score interaction profiles at the very 3' end of the $Igk$ locus (Supplementary Fig. 7b). This gap indicates a relatively high degree of contact insulation between the $V_K$ gene region and the "regulatory" loop, which contains the $J_K$, $C_K$, and $Igk$ enhancer elements and is likely formed between the upstream Sis element and first downstream CBE at the $Igk$ 3' end (Fig. 5d and Supplementary Fig. 1a).

Our finding that the pre-B cells with their elevated Wapl expression level can only form loops with a median size of 250 kb (Fig. 3e) raises the question of how to explain the long-range interactions from the Cer region across the entire 3.2-Mb long $V_K$ gene cluster. In this context, it is important to note that continuous loop extrusion can lead to the collision of loops[14,47]. Hence, the folding of the large $V_K$ gene region into multiple different loops likely leads to the collision of cohesin rings at the base of these loops (Fig. 5d). We, therefore, hypothesize that the collision of loops results in the formation of a transient interaction zone that juxtaposes DNA sequences at the base of these loops next to the DNA sequences of the Cer region. This, in turn, facilitates crosslinking of these DNA sequences, thus defining specific interactions along the stripe originating from the Cer region in the Micro-C data (Fig. 5d). Due to high Wapl expression in pre-B cells, loops constantly turn over so that new loops present different DNA sequences in the interaction zone (see Supplementary Movie 1), which results in a contiguous stripe consisting of all possible interactions along the $V_K$ gene cluster that can be detected in the large population of one million pre-B cells analyzed. Hence, this model could explain how contraction of the $Igk$ locus through the formation of multiple intervening loops can bring a 5' distant $V_K$ gene near the loop base into close vicinity of the 3' proximal $J_K$ elements to facilitate $V_K$-$J_K$ recombination in pre-B cells (Fig. 7).

## High density of long-range interactions across the entire $V_H$ gene cluster in pro-B cells

The Micro-C pattern at the $Igh$ locus in $Rag2^{-/-}$ pro-B cells was dominated by two strong stripes emanating from the IGCR1 and 3' CBEs (Fig. 6a and Supplementary Fig. 8a), which are known to act as loop anchors for long-range interactions across the entire $Igh$ locus[20,30]. Notably, there was a high and relatively uniform density of long-range interactions across the entire $V_H$ gene cluster in pro-B cells, while only weak substructures could be detected (Fig. 6a) in marked contrast to the situation observed at the $Igk$ locus in pre-B cells (Fig. 5b). The high density of interactions is best explained by the presence of 125 forward-

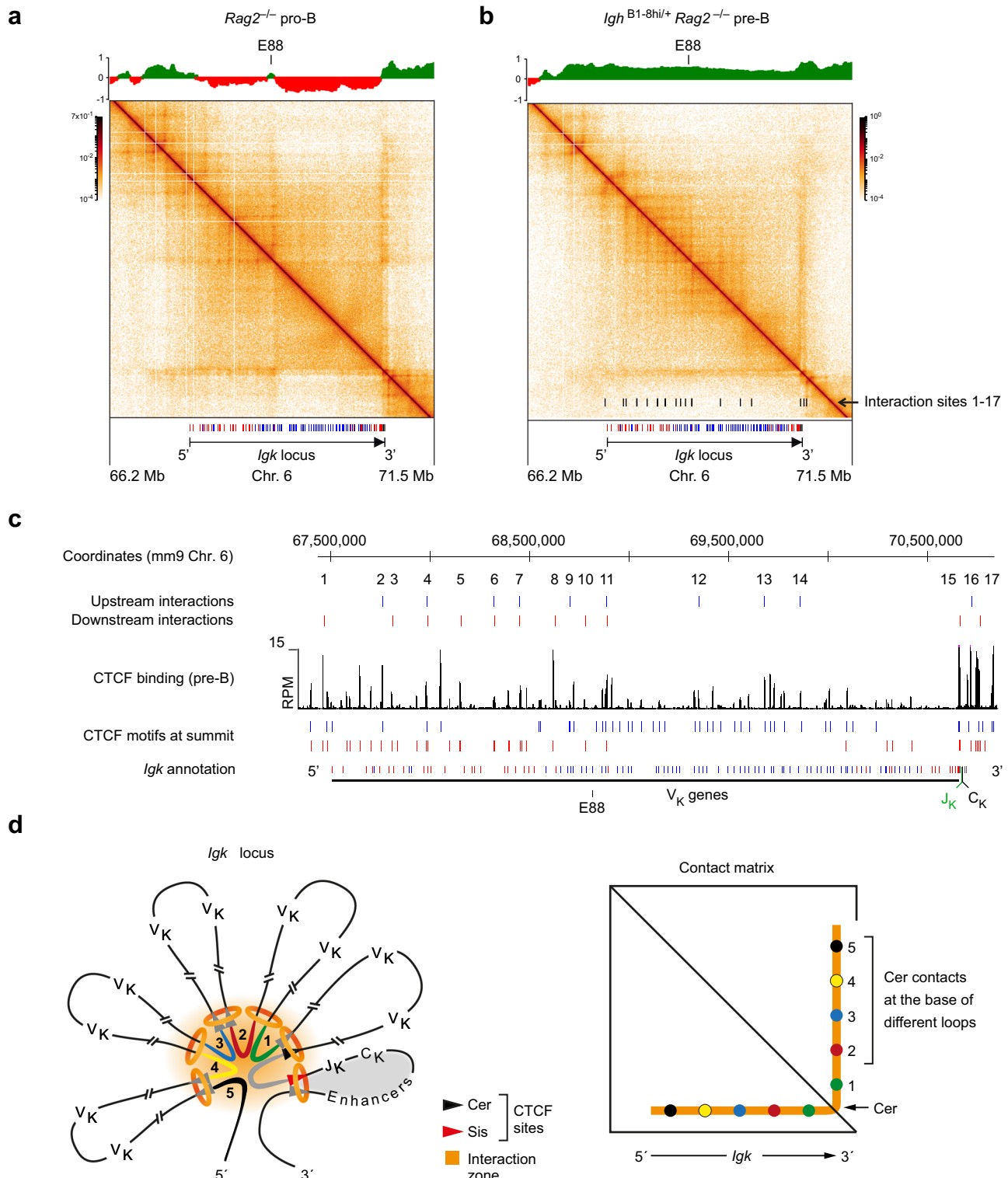

oriented CBEs in the $V_H$ gene cluster and reverse-oriented CBEs at the IGCR1 and 3'CBE elements at the 3' end of the *Igh* locus (Fig. 6b), which facilitate loop extrusion across the entire *Igh* locus in pro-B cells[20]. Loop extrusion likely initiates at random positions in the $V_H$ gene cluster and initially proceeds in a symmetrical manner, until the cohesin ring interacts with a CTCF protein bound to the next upstream forward CBE, which leads to stabilized binding of cohesin at this site[48]. Thereafter, asymmetric loop extrusion reels the DNA of the downstream *Igh* regions into the loop, until it is halted by a CTCF protein bound to a reverse CBE in convergent orientation at the IGCR1 or 3'CBE

elements[5,49]. As predicted by this extended loop extrusion model, all the different sequences of the *Igh* locus should transiently interact and thus be cross-linked during loop extrusion in individual cells of the large pro-B cell population analyzed, which likely explains the observed high density of long-range interactions across the entire $V_H$ gene cluster.

### Loss of long-range loops at the *Igh* locus due to increased Wapl expression in pre-B cells

As pro-B and pre-B cells significantly differ in their chromosomal architectures, we next analyzed the interaction pattern at the *Igh* locus

**Fig. 5 | High-resolution mapping of loop interactions at the *Igk* locus in Rag2-deficient pro-B and pre-B cells. a, b** Contact matrices of the *Igk* region (on chromosome 6) in ex vivo sorted *Rag2*[−/−] pro-B cells (**a**) and *Igh*[B1-8hi/+] *Rag2*[−/−] pre-B cells (**b**). The interaction data were determined by Micro-C analysis[45,46] and are displayed at a 10-kb bin resolution with the HiGlass visualization tool[81]. Each dot on the contact matrix represents the contact intensity between a pair of nucleosomes according to the density scale shown. White lines denote regions that could not be mapped due to too low contact density. The annotation and orientation of the *Igk* locus are shown below. The PC1 (eigenvector) values, which define the compartments A (green) and B (red), are shown above the contact matrices together with the location of the E88 enhancer[42]. The locations of the contact sites that generate the different stripes at the *Igk* locus are indicated as black bars at the bottom of the contact matrix of *Igh*[B1-8hi/+] *Rag2*[−/−] pre-B cells (**b**). **c** Location of the mapped loop anchors at the *Igk* locus in *Igh*[B1-8hi/+] *Rag2*[−/−] pre-B cells. The anchor sites of loops facing upstream (blue) or downstream (red) in the *Igk* locus of pre-B cells were determined with the Cross-score program, as described in detail in Supplementary

Fig. 7, and are shown above the CTCF ChIP-seq track and the annotation of the forward (red)- and reverse (blue)-oriented CBEs. **d** Schematic diagram explaining the looping organization of the *Igk* locus. Due to the presence of forward and reverse CBEs along the V_K gene cluster, multiple different loops are formed, which leads to the collision of cohesin rings (orange) in response to ongoing loop extrusion. As a consequence, a transient interaction zone (orange) is formed that juxtaposes DNA sequences (1–5) at the base of these loops next to DNA sequences (gray) of the Cer region, which facilitates their crosslinking and defines specific interactions along the stripe emanating from the Cer region in the Micro-C data. Due to the high Wapl expression in pre-B cells, loops constantly turn over so that new loops present different DNA sequences in the interaction zone (see Supplementary Movie 1), which results in a contiguous stripe consisting of all possible interactions along the V_K gene cluster. The orientation of CBEs in the V_K gene region is indicated by gray arrowheads, while the CBEs at Cer and Sis are shown in black and red, respectively. The relatively stable "regulatory" loop containing the J_K, C_K, and *Igk* enhancer elements is indicated by gray shading.

in *Igh*[B1-8hi/+] *Rag2*[−/−] pre-B cells by visual inspection of Micro-C and Hi-C analyses (Fig. 6c and Supplementary Fig. 8b–d). Interestingly, the long-range interactions from the IGCR1 element were lost in these pre-B cells. Moreover, the interactions from the 3′CBE region were also strongly reduced throughout the V_H gene cluster in *Igh*[B1-8hi/+] *Rag2*[−/−] pre-B cells but were still efficiently formed up to the position of the V_H5-6 gene (Fig. 6c and Supplementary Fig. 8b, d) similar to the *Igh* interaction pattern observed in *Wapl*[ΔP1,2/ΔP1,2] pro-B cells[20]. In the absence of long-range interactions, multiple substructures suggestive of internal looping were observed at the V_H gene cluster in *Igh*[B1-8hi/+] *Rag2*[−/−] pre-B cells in contrast to *Rag2*[−/−] pro-B cells (Fig. 6a, c and Supplementary Fig. 8a–d). We, therefore, conclude that extended loop extrusion across the entire *Igh* locus in pro-B cells largely suppresses the formation of internal loops within the V_H gene region.

By analyzing immature B cells of an Igμ-transgenic mouse strain, we previously demonstrated that only the most 3′ proximal V_H genes of the *Igh* locus escape allelic exclusion in pre-B cells[28]. As a similar situation may exist in *Igh*[B1-8hi/+] pre-B cells, we tested this hypothesis by analyzing the V_H-DJ_H recombination pattern in immature B cells from the bone marrow of *Igh*[B1-8hi/+] mice by VDJ-seq analysis. As shown in Supplementary Fig. 8e, the majority of immature *Igh*[B1-8hi/+] B cells expressed IgM[a] from the *Igh*[B1-8hi] allele (of 129 origin), while a minor fraction expressed IgM[b] from the *Igh*[+] allele (of C57BL/6 origin), possibly due to recombination-mediated inactivation of the *Igh*[B1-8hi] gene in early B cell development[50,51]. VDJ-seq analysis of sorted immature IgM[a] B cells from *Igh*[B1-8hi/+] mice revealed that the wild-type *Igh*[+] allele gave rise to efficient recombination of only the six most 3′ proximal V_H genes up to the V_H5-6 gene (Fig. 6d). Notably, the recombination pattern of these six V_H genes in immature IgM[a] *Igh*[B1-8hi/+] B cells strongly resembled that of *Wapl*[ΔP1,2/ΔP1,2] pro-B cells[20] (Fig. 6d). Moreover, the recombination frequency of the V_H5-2 (V_H81X) gene was similar in IgM[a] *Igh*[B1-8hi/+] B cells (1.5% for 1 *Igh*[+] allele) and *Wapl*[ΔP1,2/ΔP1,2] pro-B cells (3.5% for 2 *Igh*[+] allele) (Fig. 6d).

Together, these data indicate that the increased Wapl expression in pre-B cells causes the loss of long-range interactions across the V_H gene cluster, which likely explains the previously described decontraction of the *Igh* locus, recombination of only the 6 most 3′ proximal V_H genes and allelic exclusion of all other V_H genes in pre-B cells[28].

## Discussion

How the *Igk* locus undergoes V_K-J_K recombination is still poorly understood, as its organization fundamentally differs from that of the other three antigen receptor loci. The V genes and their associated CBEs at the *Igh*, *Tcrb* (T cell receptor β), and *Tcra/d* (T cell receptor α/δ) loci are oriented in the same forward direction, while the CBEs in their 3′ proximal domain are present in reverse orientation[25], which is compatible with loop extrusion across the entire locus[20]. In contrast, about half of all V_K genes and CBEs are present in reverse orientation in

the *Igk* locus, which results in inversional V_K-J_K recombination at a high frequency. Moreover, only the *Igk* locus undergoes RAG-mediated recombination between V genes, which additionally shapes the V_K repertoire by deletion of the intervening V_K genes[52]. Here, we have shown that V_K-J_K recombination in pro-B and pre-B cells is insensitive to high Wapl expression in marked contrast to V_H-DJ_H recombination at the *Igh* locus in pro-B cells[20], which is consistent with a recent report indicating that V_K-J_K recombination is also minimally affected upon Wapl degradation in in vitro cultured v-Abl-immortalized pre-B cell lines[32]. Our Hi-C and Micro-C analyses furthermore demonstrated that the contraction of the *Igk* locus is brought about by the formation of multiple internal loops within the V_K gene cluster, which juxtaposes distal V_K genes and proximal J_K elements next to each other. Based on these data, we propose that the contraction of the *Igk* locus by many internal loops promotes diffusion-mediated alignment of the RSS sequences of distal V_K genes and proximal J_K elements to facilitate RAG-mediated cleavage and recombination (Fig. 7).

Unexpectedly, we discovered that pro-B and pre-B cells strongly differ in their chromosomal architectures with regard to their compartment structures and loop formation, although both cell types express the same low level of *Wapl* mRNA[20] (Supplementary Fig. 4e). The observed 2.2-fold increase of Wapl protein expression in pre-B cells, possibly due to enhanced protein translation or stabilization, likely causes these global changes in chromosomal architecture, consistent with our previous finding that a 1.7- to 1.9-fold increase of *Wapl* expression in *Wapl*[ΔP1/ΔP1] and *Wapl*[ΔP1,2/+] pro-B cells, respectively, abolishes V_H gene recombination across the *Igh* locus due to drastic architectural changes[20]. Interestingly, a further threefold increase of Wapl protein expression in *Wapl*[ΔP1,2/ΔP1,2] pre-B cells had only a minimal effect on the chromosomal organization (Supplementary Fig. 4h). Hence, the entire chromosomal architecture is exquisitely sensitive to a small change of Wapl protein expression during the pro-B-to-pre-B cell transition, but thereafter is quite insensitive to any further increase in Wapl concentration.

The exquisite Wapl dosage dependence of the chromosomal architecture is contrasted by the insensitivity of V_K-J_K recombination to Wapl protein changes in pro-B and pre-B cells. V_K rearrangements are known to efficiently occur across the entire 3.2-Mb *Igk* locus in pre-B cells, although these cells are only able to form chromatin loops with a medium size of 0.25 Mb, which rules out prolonged loop extrusion as a basis for V_K-J_K recombination. Instead, we show here by high-resolution Micro-C analysis that the presence of many reverse CBEs is responsible for folding the V_K gene region into multiple internal loops that are formed between convergent forward and reverse CBEs. The distance shortening induced by the multiple loops likely accounts for the previously observed contraction of the *Igk* locus in pre-B cells[28]. This folding principle invariably leads to the collision of loops[14,47] that likely results in the formation of a transient interaction zone, where

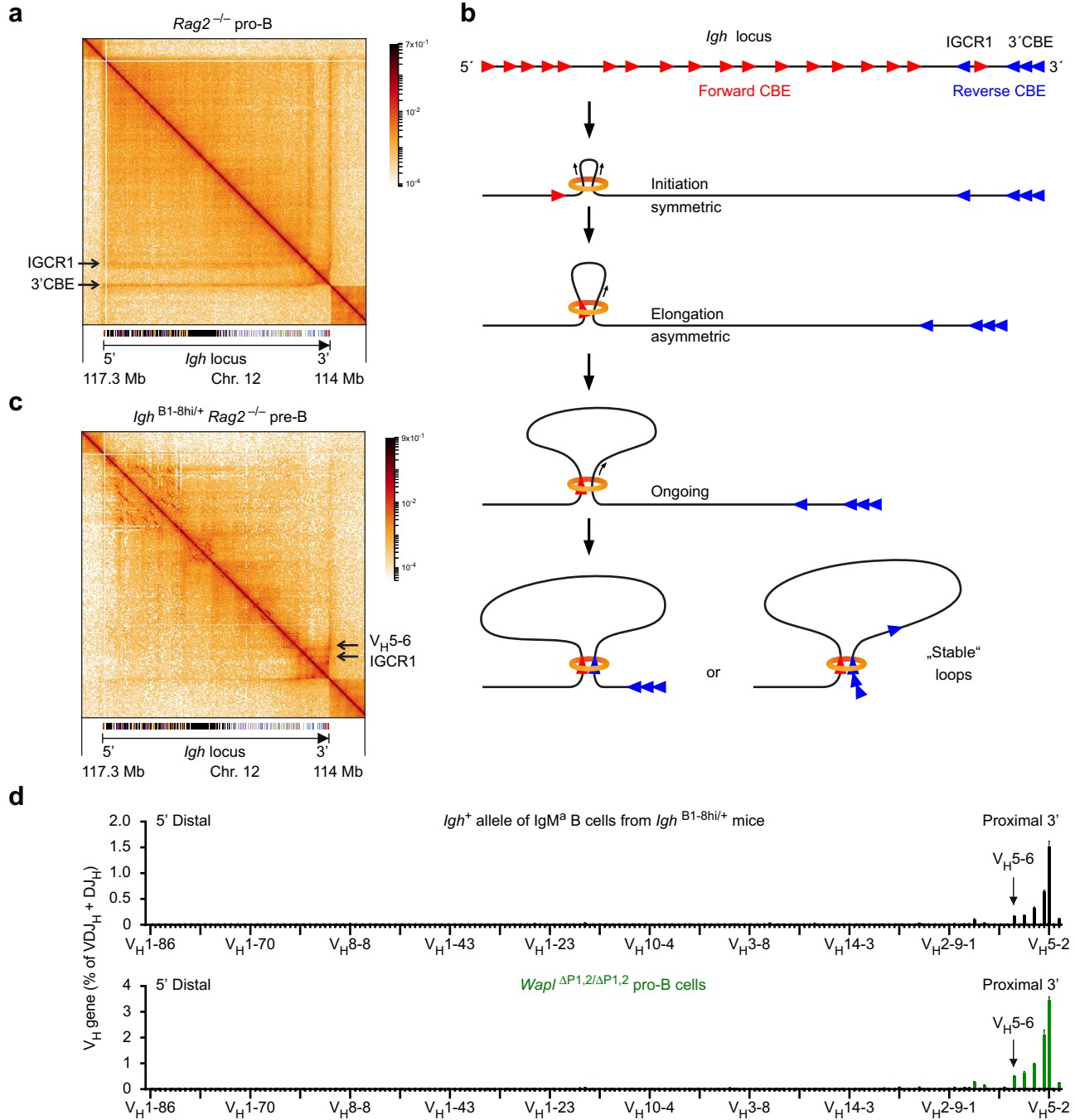

**Fig. 6 | High density of interactions across the V_H gene cluster in pro-B cells and decontraction of the *Igh* locus in pre-B cells. a** Contact matrix of the *Igh* region (on chromosome 12) in ex vivo sorted *Rag2*⁻/⁻ pro-B cells. The interaction data were generated by Micro-C analysis and are displayed at a 10-kb bin resolution with the HiGlass visualization tool[81]. Each dot on the contact matrix represents the contact intensity between a pair of nucleosomes, as displayed according to the density scale shown. White lines denote regions that could not be mapped due to low contact density. The annotation and orientation of the *Igh* locus is shown below. The stripes corresponding to interactions from the 3′CBE or IGCR1 region are indicated. **b** Schematic diagram explaining the chromatin looping at the *Igh* locus in pro-B cells. The forward CBEs (red arrowheads) in the V_H gene region are shown together with the reverse CBEs (blue arrowheads) at the IGCR1 and 3′CBE elements. Loop extrusion, upon random initiation in the V_H gene cluster, proceeds first in a symmetrical manner, until the cohesin ring (orange) interacts with a CTCF protein bound to the next upstream forward CBE, which leads to stabilized binding of cohesin at this site[48]. Thereafter, asymmetric loop extrusion reels the DNA of the downstream *Igh* regions into the loop, until it is halted by a CTCF protein bound to a

reverse CBE in convergent orientation at the IGCR1 or 3′CBE elements[5]. **c** Contact matrix of the *Igh* region in ex vivo sorted *Igh*^B1-8hi/+ *Rag2*⁻/⁻ pre-B cells. See **a** for an explanation. The position of the IGCR1 element is indicated together with the location of the V_H5-6 gene. **d** *Upper part*: VDJ-seq data obtained with sorted immature IgM^a B cells from *Igh*^B1-8hi/+ (black) mice (Supplementary Fig. 8e). Only the V_H gene reads originating from the *Igh*⁺ allele are shown, as the reads of the 5′ distal V_H1-72 (B1-8hi) gene originating from the *Igh*^B1-8hi allele were eliminated together with the reads that mapped to the related V_H1−53 gene exhibiting high sequence similarity to the V_H1-72 gene. The recombination frequency of each V_H gene is indicated as a percentage of all VDJ_H and DJ_H rearrangements and is shown as a mean value with SEM based on eight independent VDJ-seq experiments. The different V_H genes (horizontal axis) are aligned according to their position in the *Igh* locus[25] (Supplementary Data 1b). **d** Lower part: The published V_H gene recombination pattern of *Wapl*^ΔP1,2/ΔP1,2 pro-B cells[20] is shown as reference data in green. The recombination frequency of the V_H5-2 (V_H81X) gene was 1.5% (1 *Igh*⁺ allele) in IgM^a *Igh*^B1-8hi/+ B cells and 3.5% (2 *Igh*⁺ alleles) in *Wapl*^ΔP1,2/ΔP1,2 pro-B cells. Source data are provided in the Source Data file.

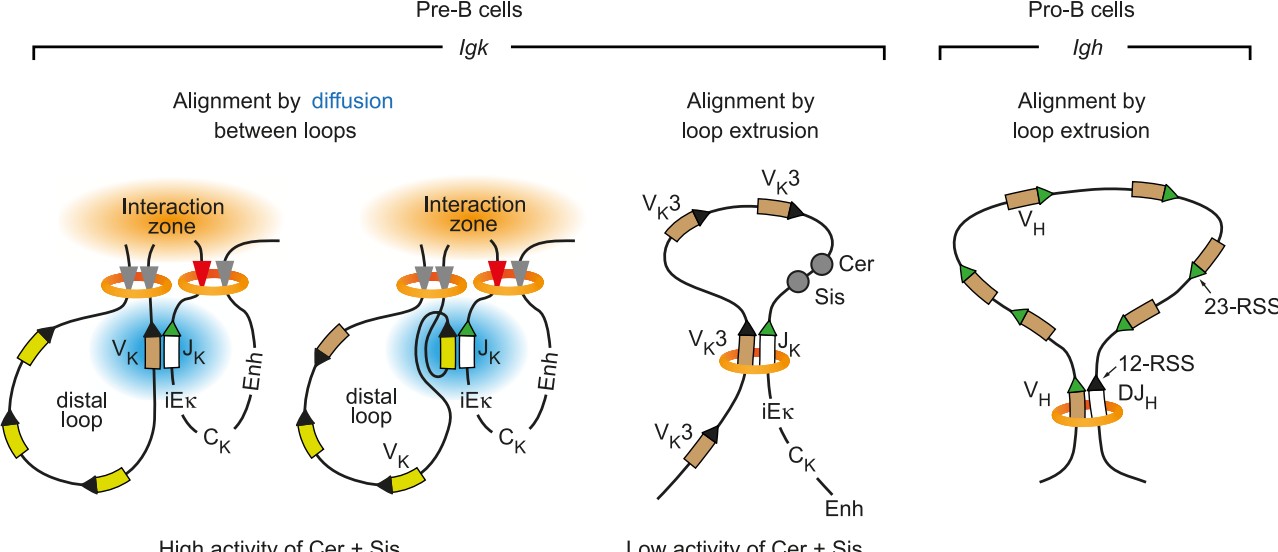

**Fig. 7 | Different folding principles leading to convergent alignment of RSS sequences at the *Igk* and *Igh* loci.** Schematic diagrams depict the different folding principles leading to a convergent alignment of RSS sequences during V gene recombination at the *Igk* and *Igh* loci. The convergent orientation of the 12-RSS (recognition signal sequence with a 12-bp spacer, black arrowhead) of the $V_K$ genes or $D_H$ segment and the 23-RSS (with a 23-bp spacer, green arrowhead) of the $J_K$ element or $V_H$ genes is essential for mediating RAG-dependent cleavage and recombination[21]. Cohesin (orange)-mediated loop extrusion across the entire *Igh* locus is responsible for the convergent alignment of the RSS sequences of $V_H$ genes and the $DJ_H$-rearranged segment prior to RAG-mediated cleavage and recombination in pro-B cells (right)[20]. Similarly, the forward-oriented members of the most 3′ proximal $V_K3$ gene family could be convergently aligned with a $J_K$ element by loop extrusion under conditions where the insulating activity of Cer and Sis is reduced or lost (middle). All other $V_K$ genes can undergo convergent alignment with a $J_K$ element only by local diffusion[62] (blue), once these elements are brought into close proximity at the interaction zone (orange), which is generated by the collision of multiple loops leading to contraction of the $V_K$ gene region (left). Enh, 3′Eκ and Edκ enhancers.

distant DNA sequences at the base of the loops are juxtaposed next to proximal DNA sequences of the Cer region, thus explaining the observed long-range interactions across the *Igk* locus in pre-B cells (Fig. 5d). $V_K$-$J_K$ rearrangements are known to occur at a low frequency in pro-B cells[34,41] due to STAT5-mediated suppression of *Igk* activation in response to IL-7 signaling in pro-B cells[53,54]. Here, we have shown that the long-range interaction pattern and compartment structure at the *Igk* locus also differ between pro-B and pre-B cells. While the entire *Igk* locus is encompassed by the transcriptionally active compartment A in pre-B cells, it is present in the less accessible compartment B in pro-B cells. Compartment B is interrupted only by a small stretch of compartment A located at the B cell-specific E88 enhancer that is known to primarily activate the recombination of adjacent $V_K$ genes in pro-B and pre-B cells[42]. The *Igk* locus contains only low levels of the poised histone mark H3K4me1 in pro-B cells in marked contrast to pre-B cells[36], which is consistent with the presence of compartment B in pro-B cells and compartment A in pre-B cells.

The two elements Cer and Sis in the $V_K$-$J_K$ intervening region play an important role in promoting the recombination of central and distal $V_K$ genes by suppressing excessive usage of the 3′ proximal $V_K$ genes[55–57]. The Cer element is essential for the contraction of the *Igk* locus[56] and contains two reverse-oriented CBEs that facilitate interactions across the entire $V_K$ gene region (Fig. 5d), while the inversion of these CBEs promotes the recombination of proximal $V_K$ genes at the expense of central and distal $V_K$ genes[58]. The Sis element, whose deletion results in a less prominent overactivation of proximal $V_K$ gene recombination[55], contains two forward-oriented CBEs[25] (Supplementary Fig. 1a) that are able to participate in the formation of the "regulatory" loop containing the $J_K$, $C_K$, and *Igk* enhancer elements (Fig. 5d). Notably, the function of Cer and Sis appears to be only essential for establishing a balanced $V_K$ gene usage across the $V_K$ gene cluster during the primary $V_K$-$J_K1$ recombination event, which simultaneously leads to the deletion or inversional 5′ translocation of these two elements. Notably, conditional loss of CTCF in pre-B cells increases the interactions of the 3′ proximal $V_K$ genes with the downstream *Igk* enhancers, which strongly promotes recombination of these proximal $V_K$ genes at the expense of central and distal $V_K$ genes[59]. Furthermore, a similar phenotype was seen upon double deletion of Cer and Sis[57]. Here, we also observed increased recombination of the 3′ proximal $V_K3$ gene family in *Wapl*$^{ΔP1,2/ΔP1,2}$ versus *Wapl*$^{+/+}$ pre-B cells as well as in *Wapl*$^{ΔP1,2/ΔP1,2}$ versus *Wapl*$^{+/+}$ pro-B cells. Hence, the decreased residence time of cohesin on chromatin upon increased Wapl expression in *Wapl*$^{ΔP1,2/ΔP1,2}$ cells may interfere with CTCF-mediated insulation of the 3′ proximal $V_K$ genes by Cer and Sis, thus leading to enhanced interactions of the proximal $V_K3$ genes with the downstream *Igk* enhancers.

The orientation of the RSS sequences at the V, D, and J elements of all antigen receptor genes requires that they are convergently aligned in the 3′ proximal RAG+ recombination center prior to RAG-mediated cleavage and recombination[33]. Detailed molecular analysis of $D_H$-$J_H$ recombination[60] and $V_H$-$DJ_H$ recombination[3,20,61] at the *Igh* locus has provided strong evidence for the convergent alignment of RSS sequences by loop extrusion (Fig. 7). Convergent alignment by loop extrusion prior to $V_H$-$DJ_H$ recombination requires, however, that all $V_H$ genes of the *Igh* locus are present in the same forward orientation, as a $V_H$ gene upon its inversion fails to recombine in pro-B cells[20]. As half of the $V_K$ genes are present in reverse orientation in the *Igk* locus, convergent RSS alignment by loop extrusion is impossible except for the forward-oriented members of the most 3′ proximal $V_K3$ gene family, which may only be possible upon loss of the insulating activity of Cer and Sis (Fig. 7). We, therefore, hypothesize that the RSS sequences of $V_K$ and $J_K$ elements, which are brought into close proximity in the interaction zone by contraction of the $V_K$ gene region through multiple loops, are aligned by local diffusion[62] prior to $V_K$-$J_K$ recombination (Fig. 7).

We have previously demonstrated that the non-functionally rearranged *Igh* allele undergoes decontraction in response to pre-BCR signaling, which results in feedback inhibition of $V_H$-$DJ_H$ recombination except for the most 3′ proximal $V_H$ genes that escape allelic

exclusion in pre-B cells[28]. However, the molecular mechanism causing *Igh* decontraction has remained elusive until to date. Here, we have shown by high-resolution Micro-C analysis that the long-range interactions from the IGCR1 region are lost in *Igh*[B1-8hi/+] *Rag2*[−/−] pre-B cells. Moreover, the interactions from the 3′ CBE region are strongly reduced beyond the location of the $V_H$5-6 gene, similar to the interaction pattern observed in *Wapl*[ΔP1,2/ΔP1,2] pro-B cells[20]. Notably, the loss of extended loop extrusion at the *Igh* locus in *Igh*[B1-8hi/+] pre-B cells led to recombination of only the six most 3′ proximal $V_H$ genes up to the $V_H$5-6 gene and possibly to allelic exclusion of all other $V_H$ genes, which strongly resembles the $V_H$ gene recombination pattern of *Wapl*[ΔP1,2/ΔP1,2] pro-B cells[20]. These data, therefore, suggest that the increased Wapl expression in pre-B cells may be the molecular cause for *Igh* decontraction and allelic exclusion in pre-B cells. Future genetic experiments aiming at the downregulation of Wapl expression in pre-B cells will be required to conclusively demonstrate an essential role of Wapl in the control of allelic exclusion at the *Igh* locus.

In summary, we have shown that pro-B and pre-B cells have distinct chromosomal architectures and that the recombination of V genes at the equally large *Igh* and *Igk* loci is facilitated by fundamentally different folding principles in pro-B and pre-B cells, respectively. As decontraction and allelic exclusion of the non-functionally rearranged *Igh* allele likely depend on increased Wapl expression in pre-B cells, it could be argued that the *Igk* locus had to assume a different organization and folding principle to undergo efficient $V_K$-$J_K$ recombination under conditions of high Wapl expression.

## Methods

### Mice
The following mice were maintained on the C57BL/6 background: *Wapl*[ΔP1,2/ΔP1,2] mice[20], *Rag2*[−/−] mice[63], and *Igh*[B1-8hi/B1-8hi] mice[43]. Experimental and control mice were co-housed under standard pathogen-free conditions at a temperature of 22 °C and 55% humidity with a day cycle of 14 h light and 10 h dark and with unrestricted access to food and water. Cells were harvested from mice that were 4–5-week-old (VDJ-seq analysis), 5–6-week-old (Hi-C and Micro-C analysis), and 4–6-week-old (immunoblot analysis). Mice were euthanized by carbon dioxide inhalation. Both female and male mice were used at a similar ratio in this study. All mouse experiments were carried out according to valid project licenses, which were approved and regularly controlled by the Austrian Veterinary Authorities.

### Antibodies and flow-cytometric analysis
The following monoclonal antibodies were used for flow-cytometric analysis of mouse bone marrow cells: B220/CD45R (RA3-6B2; BD; 1:200), CD19 (1D3; BD; 1:300), CD25/IL-2Rα (PC61; BD Pharmingen; 1:500), CD117/Kit (2B8; Invitrogen; 1:1000), IgD (11-26c, Invitrogen; 1:2000), IgM (II/41, Invitrogen; 1:300), IgMa (MA-69, BioLegend; 1:1000), and IgMb (AF6-78, BioLegend; 1:1000). The following antibodies were used for immunoblot or immuno-precipitation analyses: anti-Wapl (rabbit polyclonal Ab, A960; Peters laboratory), anti-Tbp (mouse mAb clone 3TF1-3G3; Active Motif), and anti-H3K27ac (rabbit polyclonal Ab, ab4729; Abcam).

B cell types in the bone marrow were defined as CD19[+]B220[+]IgM[−]IgD[−]Kit[+]CD25[−] pro-B cells, CD19[+]B220[+]IgM[−]IgD[−]Kit[−]CD25[+] pre-B cells, and CD19[+]B220[+]IgMa[g]IgD[−]Kit[−] immature B cells. Flow-cytometric experiments and cell sorting were performed on LSR Fortessa (BD Biosciences) and FACSAria III (BD Biosciences) machines, respectively, using the FACS Diva (8.0) software. Flowjo software (Treestar) was used for data analysis.

### Protein extract preparation and immunoblot analysis
Ex vivo pro-B and pre-B cells were sorted from the bone marrow by flow cytometry, and whole-cell extracts were prepared, using 2x SDS-PAGE sample buffer containing β-mercaptoethanol. The proteins were denatured by boiling, separated by SDS-PAGE, and analyzed by immunoblot analysis. The signal intensity of protein bands was quantified using ImageJ software and normalized to that of the Tbp loading control.

### VDJ-seq analysis
VDJ-seq analysis of recombination at the *Igk* and *Igh* loci was performed as described in ref. 37. Genomic DNA was extracted from ex vivo sorted pro-B, pre-B, and immature B cells. The DNA (2 μg) was sheared using the Bioruptor sonicator (Diagenode) and subjected to end-repair and A-tailing, followed by ligation of adapters containing 12 UMI sequences using the NEBNext Ultra II DNA library prep kit for Illumina (NEB). A primer extension step with biotinylated $J_K$- or $J_H$-specific primers generated the single-stranded DNA products that were captured using Dynabeads MyOne streptavidin T1 beads (Thermo Fisher Scientific) and PCR-amplified with nested $J_K$- or $J_H$-specific and adapter-binding primers[37]. The Illumina sequencing adapter primers, including the indexes for multiplexing of libraries, were added to the PCR products in a final PCR amplification step. Paired-end 300-bp sequencing was performed on a MiSeq (Illumina) sequencing instrument (Supplementary Data 3). The bioinformatic analysis of the VDJ-seq data was performed as described in detail[37], and the resulting data were processed for display in the respective figures using R version 3.3.3.

### cDNA preparation for RNA-sequencing
Total RNA from ex vivo sorted pre-B cells was isolated with the RNeasy Plus Mini Kit (Qiagen), and mRNA was purified by two rounds of poly(A) selection with the Dynabeads mRNA purification kit (Invitrogen). The mRNA was fragmented by heating at 94 °C for 3 min in a fragmentation buffer and cDNA was prepared as described in ref. 20.

### Library preparation and Illumina deep sequencing
About 0.6–20 ng of cDNA or ChIP-precipitated DNA was used as starting material for the generation of sequencing libraries with the NEBNext Ultra II DNA library prep kit for Illumina (NEB). Alternatively, sequencing libraries were generated using the NEBNext End Repair/dA-Tailing Module and NEBNext Ultra Ligation Module (NEB) followed by amplification with the KAPA Real-Time Amplification kit (KAPA Biosystems). Cluster generation and sequencing were carried out using the Illumina HiSeq 2500 system with a read length of 50 nucleotides, according to the manufacturer's guidelines.

### Hi-C library preparation
*Wapl*[+/+], *Wapl*[ΔP1,2/ΔP1,2], and *Igh*[B1-8hi/+] *Rag2*[−/−] pre-B cells were isolated from the bone marrow by immunomagnetic enrichment with anti-CD19-MicroBeads (Miltenyi Biotec) and were subsequently sorted by flow cytometry as CD19[+]B220[+]IgM[−]IgD[−]Kit[−]CD25[+] pre-B cells prior to Hi-C library preparation. Hi-C libraries were prepared from $2 × 10^7$ cells as described in detail in ref. 5 and were sequenced using the Illumina NextSeq system with a read length of 75 nucleotides in the paired-end mode, according to the manufacturer's guidelines.

### Micro-C library preparation
Pro-B cells from the bone marrow of *Rag2*[−/−] mice were isolated by immunomagnetic enrichment with anti-CD19-MicroBeads (Miltenyi Biotec) followed by flow-cytometric sorting as CD19[+]B220[+]IgM[−]IgD[−]Kit[+]CD25[−] cells, while *Igh*[B1-8hi/+] *Rag2*[−/−] pre-B cells were sorted as CD19[+]B220[+]IgM[−]IgD[−]Kit[−]CD25[+] pre-B cells. Micro-C libraries[45] were prepared from $1 × 10^6$ cells using the Dovetail Micro-C Kit (# 21006) according to the manufacturer's user manual (https://dovetailgenomics.com/wp-content/uploads/2021/09/Dovetail%E2%84%A2-Micro-C-Kit-User-Guide-Version-1.2.pdf). Libraries were sequenced using the NovaSeq 6000 S4 system with

a read length of 150 nucleotides in the paired-end mode, according to the manufacturer's guidelines.

## Bioinformatic analysis of CTCF peaks in the *Igk* locus

We identified CTCF peaks (here referred to as CTCF-binding elements; CBEs) in the *Igk* locus based on the published data of our CTCF antibody ChIP-seq experiment (GSM1145865) that was performed with short-term cultured *Rag2*[−/−] pro-B cells[30]. In addition, published CTCF ChIP-seq data of ex vivo sorted pre-B cells[36] (GSM2973687) were used. Sequence reads were uniquely aligned to the mouse genome assembly version of July 2007 (NCBI37/mm9) using the Bowtie program version 1.0 (ref. 64). CTCF peaks were called by MACS 2.2.5 (ref. 65) and filtered for *P* values of $<10^{-10}$ to obtain a total of 61,354 peaks in *Rag2*[−/−] pro-B cells and 36,225 peaks in pre-B cells. For the *Igk* locus (mm9 Chr. 6; 67,505,630-70,694,944), this resulted in a total of 71 peaks in pro-B and pre-B cells, with 48 common peaks, and 19 and 4 unique peaks in pro-B and pre-B cells, respectively. We subsequently split all 71 peaks using PeakSplitter[66] to obtain a final list of peak summits. This resulted in 112 CBEs in pro-B cells and 53 CBEs in pre-B cells across the *Igk* locus (Supplementary Fig. 1a).

To enumerate all potential CTCF-binding sites in the *Igk* locus, we retrieved the repeat-masked mouse genome sequence (mm9) using EXONERATE[67] and scanned the sequence region of the *Igk* locus with a CTCF motif derived from the summits of the top 300 CTCF peaks, using MEME[68]. The scanning was done with FIMO version 4.9.1 (ref. 69) by setting the *P* value threshold to <0.001, which resulted in 77 motifs in pro-B cells and 52 motifs in pre-B cells that were clearly assigned to a CBE within 100 bp of the peak summit (shown in Supplementary Fig. 1a). In case of ambiguity, we selected the motif with the higher score.

## Analysis of RNA-seq data

The number of reads per gene was counted using the featureCounts version 1.5.0 (ref. 70) with default settings. Transcripts per million (TPM) values were calculated as described[71]. Differential gene expression between ex vivo sorted *Wapl*[+/+] and *Wapl*[ΔP1,2/ΔP1,2] pre-B cells (Fig. 2f and Supplementary Data 2) was analyzed using R version 3.3.3. and DESeq2 version 2.1.14.1. Regularized log transformations were computed with the blind option set to "FALSE". Genes with an adjusted *P* value of <0.05, TPM (averaged for each genotype) of >5 at least in one of the two genotypes, and a fold-change of >2 were called as significantly differentially expressed. All transcripts of the V, D, and J gene segments at the *Igh*, *Igk* and *Igl* loci were eliminated from the list of significantly regulated genes, although the immunoglobulin and T cell receptor transcripts were included in all TPM calculations.

## Processing, normalization, and resolution of Hi-C data

The HiCUP pipeline version 0.5.10 (ref. 72) with the scorediff parameter set to "10" was used to truncate, align and filter the reads by applying the following software versions: R 3.4.1 (https://www.r-project.org), Bowtie 2.2.9 (ref. 64), and SAMtools 1.4. (ref. 73). Contact matrix files have been produced with the Juicer tools 1.8.9 (ref. 74). The resolution of the Hi-C data has been calculated according to ref. 5 by using the script "calculate_map_resolution". The following unique di-tags were generated; 411,290,986 (GSM6427693) and 105,703,179 (GSM6427695) with *Wapl*[+/+] pre-B cells, 694,115,546 (GSM6427694) and 76,038,129 (GSM6427696) with *Wapl*[ΔP1,2/ΔP1,2] pre-B cells as well as 452,767,106 (GSM6427697) with *Igh*[B1-8hi/+] *Rag2*[−/−] pre-B cells. The following maximal resolution of the Hi-C data was calculated; 6.65 kb (*Wapl*[+/+] pre-B cells), 4.25 kb (*Wapl*[ΔP1,2/ΔP1,2] pre-B cells), and 11.8 kb (*Igh*[B1-8hi/+] *Rag2*[−/−] pre-B cells). We also analyzed the different Hi-C datasets with the open2c distiller-nf pipeline [https://github.com/open2c/distiller-nf], normally used for Micro-C data analysis (see below), to be able to compare Hi-C and Micro-C data with each other,

to calculate the compartmentalization scores for the saddle plot analysis (Supplementary Fig. 5b) and to analyze the contact frequencies across the *Igk* locus in detail (Supplementary Fig. 3). Both pipelines (HiCUP and distiller-nf) led to very similar contact matrices for the same cell type and genotype.

## Analysis of intrachromosomal contact frequency (Hi-C)

Contact frequency distributions have been calculated using the makeTagDirectory command of HOMER 4.10.3 (ref. 75). The contact frequency plots shown in Fig. 2a and Fig. 3b, f are based on ~50 contact data points based on ~50 bins, whereby each bin is defined as 0.1 step on the $\log_{10}$ scale of genomic distance observed between the contact points. Each data point is thus the sum of all contact fraction values in the respective bin. The contact frequency plot is shown as a smoothened line of the ~50 contact data points plotted against the logarithmic ($\log_{10}$) genomic distance. Detailed contact frequencies across the *Igk* locus (Supplementary Fig. 3), as well as the saddle plots (Supplementary Fig. 5b), were analyzed and generated with Cooltools[76] and Python 3.8.13 (ref. 77), based on the data obtained with the open2c distiller-nf pipeline. To this end, we followed the description in the Cooltools notebook to generate the contact frequency [P(s)] plots, their first derivative (slope) curves, as well as the saddle plots for the various cell types and genotypes.

## Analysis of chromatin loops (Hi-C)

Intrachromosomal loops have been called with the HiCCUPS algorithm from the Juicer tools[74], based on the contact matrix derived from both Hi-C replicates of the same cell type. For the Hi-C comparisons between *Wapl*[+/+] and *Wapl*[ΔP1,2/ΔP1,2] pre-B cells (Supplementary Fig. 4b), we down-sampled the aligned reads obtained with the *Wapl*[ΔP1,2/ΔP1,2] pre-B cells to the same read number obtained with the *Wapl*[+/+] pre-B cells. In contrast, the Hi-C analyses resulted in similar read numbers for *Wapl*[+/+] pro-B and *Wapl*[+/+] pre-B cells (Supplementary Fig. 4g) as well as for *Wapl*[+/+] pre-B and *Wapl*[ΔP1,2/ΔP1,2] pro-B cells (Supplementary Fig. 5d). The obtained loops identified by HiCCUPS were overlapped between the two cell types of each cell pair to be compared, using "bedtools intersect" from the BEDTools package version 2.27.1 (ref. 78), with command parameters "-wa -wb -f 0.9 -r". A reciprocal minimal overlap of 90% was required for two loops to be called "common", while all other loops were referred to as "unique". The distributions of loop lengths were plotted with R.

## Analysis of Hi-C contact correlations

Correlation coefficients between Hi-C contact maps were calculated with HiCRep.py[79], by using the parameters "−binsize 25000 -h 10 0dBPMax 20000000 −bdownSample −excludeChr chrM chrX chrY", and by using two Hi-C contact matrices to be compared as input in the cooler format.

## Bioinformatic analysis of Micro-C data

The open2c distiller-nf pipeline [https://github.com/open2c/distiller-nf] was used to process Micro-C datasets. The pipeline essentially aligns all sequences to the reference genome (mm9), parses the alignments into interaction pairs, filters out PCR duplicates, and finally aggregates pairs into binned matrices of Micro-C interactions. The underlying methods are all based on the pairtools (https://github.com/open2c/pairtools) and cooler[80].

The stripes of enriched contacts in Micro-C maps at 2 kb resolution were detected using the Cross-score algorithm (https://github.com/glab-vbc/cross-score). Briefly, for each genomic bin, Cross-score calculates the total frequency of all contacts made with its neighbors located in a given range of distances, by separately analyzing the contacts with upstream or downstream neighbors. This score reveals genomic bins that participate in long-range interactions at a higher-than-normal frequency. Since our goal was to detect genomic bins that

anchor focal loops or contact stripes, we calculated the Cross-score values for interactions at distances between 30 kb and 1 Mb, a typical range of interactions formed by cohesin-mediated looping. Peaks in Cross-score profiles were called using the ggpmisc package (Pedro J. Aphalo, 2021; https://github.com/aphalo/ggpmisc).

Eigenvalue decomposition to calculate compartment signals was performed with the eigs-cis program from the cooltools package (https://zenodo.org/record/5214125#.YhjAXJYo_mE), using H3K27ac ChIP-seq data of ex vivo sorted *Rag1*^Cre/Cre pro-B cells (GSM6427700; Supplementary Data 3) as active chromatin phasing data. For smoother visualization of interaction and compartment tracks, we imputed missing values (due to missing mapping information) by using the imputeTS R packages (Moritz, Steffen, and Bartz-Beielstein (2017) "imputeTS: Time Series Missing Value Imputation in R." R Journal 9.1, doi: 10.32614/RJ-2017-009). All resulting files (multi-resolution cooler files, compartment scores, and interaction scores) were visualized using the HiGlass visualization tool[81] (http://higlass.io).

### Statistical analysis

Statistical analysis was performed with the GraphPad Prism 7 software. Two-tailed unpaired Student's *t*-test analysis was used to assess the statistical significance of one observed parameter between two experimental groups. The statistical evaluation of the RNA-seq data is described above (Analysis of RNA-seq data).

### Reporting summary

Further information on research design is available in the Nature Portfolio Reporting Summary linked to this article.

## Data availability

The RNA-seq, ChIP-seq, VDJ-seq, Hi-C, and Micro-C data reported in this study (Supplementary Data 3) are available at the Gene Expression Omnibus repository under the accession number GSE210289. Source data are provided with this paper.

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

## Acknowledgements

We thank Karin Aumayr's team for flow-cytometric sorting and Andreas Sommer's team at the Vienna BioCenter Core Facilities for Illumina sequencing. This research was supported by Boehringer Ingelheim, the Austrian Research Promotion Agency (FFG-878286), the European Research Council (ERC) under the European Union's Horizon 2020 research and innovation program (grant agreement No 740349 [M.B.] and No 1020558 [J.-M.P.]), the Human Frontier Science Program (grant RGP0057/2018 [J.-M.P.]), and the Vienna Science and Technology Fund (grant LS19-029 [J.-M.P.]).

## Author contributions

L.H. performed most experiments; G.W. generated the Hi-C and Micro-C data; L.C. isolated and prepared RAG2-deficient pro-B and pre-B cells for Hi-C and Micro-C analyses; H.T. performed the VDJ-seq analysis of immature *Igh*[B1-8hi/+] B cells; M.J. performed all bioinformatic analyses; J.-M.P. provided supervision and advice on cohesin biology; A.G. provided advice on loop extrusion and Micro-C analysis; L.H. and M.B. planned the project, designed the experiments, and wrote the manuscript.

## Competing interests

The authors declare no competing interests.
