## [Peer Review File · Nature Communications]

Igh and Igk loci use different folding principles for V gene recombination due to distinct chromosomal architectures of pro-B and pre-B cellsREVIEWER COMMENTS

Reviewer #1 (expertise in VDJ recombination, spatial regulation of antigen receptor loci accessibility):

Recent work by the Busslinger and Alt labs has demonstrated how Pax5- and Wapl-mediated regulation of loop extrusion at the Igh locus accounts for Igh locus contraction and capture of V gene segments for V-to-DJh recombination in pro-B cells. A similar loop extrusion scenario cannot easily explain capture of Vk gene segments for Vk-to-Jk recombination in pre-B cells, since many Igk CBEs and V segments are present in reverse orientation with respect to the recombination center. Here the authors show that unlike the Igh locus in pro-B cells, Igk locus conformation and recombination are relatively insensitive to mutational elevation of Wapl expression in pre-B cells. They attribute relative insensitivity of Igk locus and genome-wide architecture to elevated Wapl protein in pre-B relative to pro-B cells, in a manner that is sufficient to promote architectural changes in Igk, Igh, and genome-wide, with shorter loops in pre-B cells. They also note that low frequency Igk rearrangement in pro-B cells is biased towards distal Vk segments even with Wapl upregulated by mutation, conditions in which rearrangement of distal Vh gene segments is strongly disfavored. Based on these behaviors and maintenance of long distance Igk chromatin contacts under these conditions, they argue for a different mechanism of folding as compared to Igh, whereby Igk loop organization in pre-B cells provokes loop collisions that bring distantly located V segments into contact with the 3' portion of the locus. Finally, they provide evidence that the switch to shorter range looping in pre-B cells can explain Igh allelic exclusion, which affects all but the most proximal Vh genes.

This is a topic of much interest and the results are potentially significant. However, as presented, the work suffers from two major problems (outlined in greater detail below): (1) insufficient quantitative analysis, weakening the conclusions about differences and similarities in rearrangement and genomic contact profiles; and (2) failure to analyze Igk contacts in WT vs Wapl mutant in the absence of recombination; over-reliance on the suboptimal Igk contact profiles in rearranging B cell populations.

Specific concerns:

1. There is a lack of clarity on Igk locus contraction. In the introduction (p4) the authors quote ref#35 that shows that Igk contraction initiates in pro-B cells as well as their own earlier work (ref#28) showing it to initiate in pre-B cells. They restate elsewhere (p11) "pro-B ... already undergo contraction of the Igk locus". But later (p15) they seem to relate Igk contraction to the multiple loops that they detect specifically in pre-B cells. Please clarify.
2. The authors make comparative statements throughout that are neither quantified nor statistically analyzed. For example, on p7 it is noted that Vk3-4 to 3-12 rearrange more frequently in Wapl mutant compared to WT. Can this be presented in a more quantitative way? Similarly, on p12 it is noted that there is a difference in Vk rearrangement between pro- and pre-B cells. Can this be quantified? See also comments about chromatin contacts below.
3. Fig 2C is confusing. The figure legend indicates that the color scheme represents differential interactions in Wapl mutant relative to WT, and this is consistent with the red (+4) and blue (-4) legend in the figure panel. But they should not also code WT as blue and mutant as red, even though blue does represent interactions higher in WT and red, interactions higher in mutant. Pick one approach to define the color scheme.
4. Can the authors further elaborate on the nature of the changes in the contact maps (Fig 2b,c)? It looks as if the regions showing reduced interactions in mutant represent contacts between compartments A and B (white in b becomes blue in c). Is this interpretation correct, and if so, does it mean that elevated Wapl modestly increases insulation between nearby opposing compartments? Reciprocally, in Fig 3c, is lower Wapl in pro-B is associated with slight breakdown of compartmentalization.
5. The authors argue that changes in chromatin organization between pro-B and pre-B are due to a 2.2x increase in Wapl expression. I believe they only present one piece of data that moves this beyond a correlation, namely, that Wapl mutant pro-B cells display a distribution of contact frequencies that is much more similar to that of WT pre-B than WT pro-B (compare Fig 3f to Fig 3b). Given the importance of this result, it would be good to see a bit more primary data, for example the Hi-C contact matrix to compare to that in Fig 3c).
6. The description of differences in Igk contact maps for WT pre-B (Fig 1d), WT pro-B and mutant

pro-B (Fig 4b) is vague and non-quantitative. The same for the comparison of Igh to Igk in mutant pro-B (Extended 3c). In particular, the maintenance of long-distance Igk contacts in mutant pro-B cells, and a clear difference between Igh and Igk, are not as obvious as one might think from reading the text. And it is hard to understand how it is possible to meaningfully compare Igk contact maps between pro-B and pre-B since the extent of rearrangement is so different in the two compartments.

7. The micro-C analysis of Igk interactions on a R2KO background provides a much cleaner and more highly resolved look at interactions across the Igk locus. The authors should have analyzed Wapl-mutant mice on this background, thereby simplifying the manuscript and strengthening the conclusions.

8. It was noted in Fig 4A that Igk rearrangements in pro-B are weighted more distally than in pre-B. Can the authors relate quantitative differences in long-distance interactions in the micro-C experiments to this regional difference in Vk rearrangement? With this in mind, it looks as if there is a more well-defined loop organization in the distal as compared to the proximal half of the Vk region in pro-B cells. This difference is not as apparent in pre-B cells, where both distal and proximal regions are structured.

9. It is suggested (p14) that collisions at the bases of different loops creates the continuum of interactions that form the stripe emanating from Cer. It is not explicitly stated, but when the authors refer to constant turnover and formation of new loops, do they include as well the process of loop extrusion within these shorter loops as contributing to this continuum of contacts?

10. Extended data Fig 5c,d presents Hi-C comparison of Igh in WT pro-B to Rag2KO pre-B. What is the point of this comparison, which is far from ideal since pro-B but not pre-B will have undergone rearrangement? And the more valid comparison, of Rag2KO pro-B to Rag2KO pre-B, is presented in the panels immediately above. Justify or remove.

11. The authors propose that Igk contraction is a function of its multiple loop organization (p18). Can they relate this to the Igk loop organization in pro-B, in which the locus is contracted (see above) but in which loops are somewhat less well defined? Can they contrast this with Igh, which also shows internal looping in pre-B (Fig 5b), where the locus is decontracted?

Reviewer #2 (expertise in 3D genome conformation, hi-C, micro-C):

This is an important piece of work that addresses a gap in the field of 3D organization and recombination of light chain loci in B cells. As stated, the role of cohesin and loop extrusion is shown clearly for recombination at Igh and T cell receptor loci but is inconsistent with Igk. The reasonable model the authors propose here for Igk is its contraction by many internal loops facilitating RAG-mediated cleavage and recombination. I am very much in favor of seeing this paper published although I must say it is extremely hard to follow at times (at least for this reviewer). I have some comments below, which are important to address.

- The authors need to perform a proper analysis of loops/contacts that are differential between the compared conditions. This is important throughout the paper but more specifically for Figure 2d and 3d and how/if those observed patterns generalize genome-wide.

- The sentence that says "as a 2.2-fold increase of Wapl protein expression from Wapl+/+ pro-B to Wapl+/+ pre-B cells ..." missed the point that such increase PLUS the cell state difference is what causes drastic changes in comparison to within pre-B cells comparison with differences in Wapl levels. This statement has to be corrected to accurately reflect that. They are later careful and add "likely" to the last sentence of this section. Related to this, the comparison of Wapl del(P1,2) pro-B cells to Wapl WT+ pre-B cells should be done more thoroughly beyond showing Figure 3f if the claim is that the two are very similar.

- The conclusion that "increased Wapl protein expression had a smaller effect on both differential gene expression and genomic architecture in pre-B cells relative to pro-B cells" has to be supported by uniform analysis of RNA-seq data between the previous Pax5-Wapl paper and this paper. It should be the same analysis rather than a similar analysis. If that is already the case, the authors should state so.

- The lack of translation between mRNA level differences and protein-level measurements for Wapl between pro- and pre-B cells is certainly interesting and warrants further investigation. However, it is unclear to me how higher Wapl in pre-B cells may lead to longer-range interactions. This seems mainly due to normalization of the scaling curves such that pro-B cells having more <5Mb interactions translate into them having relatively less that are >5Mb. A difference map, such as the one in Fig 2c is needed to strengthen this argument visually.

- The interpretation of Figure 4b "the structures of the sub-TADs were, however, less well defined in Wapl+/+" has to be accompanied by quantification. The heatmaps are not strikingly different.

Minor

- contact frequencies -> contact frequencies

- Wapl^{high} terminology is used sporadically. I suggest they early on define Wapl del(P1,2) as Wapl^{high} and go with it. It makes it much easier to read.

- The video clip is a nice attempt to summarize their model but it is far from self-explanatory. Not sure if this can be made better by more text and/or narration, the reference to the video in the text has a good description.

Reviewer #3 (expertise in Ig loci regulation, VDJ recombination, chromatin accessibility during VDJ recombination):

This is a very interesting study that follows up on the authors' recent seminal study (Hill et al 2020) which showed that the B cell specific transcription factor, Pax5 orchestrates its widespread effects on gene regulation in progenitor B cells via a newly discovered mechanism. Specifically, it binds and represses the promoter of the WAPL cohesin release factor gene, thereby reducing WAPL expression in proB cells. That paper shows that this leads to reduced cohesin residence time on the large immunoglobulin heavy chain locus, thereby inhibiting DNA loop extrusion, shown by the authors and others to provide the underlying mechanism that brings V and D genes together in this very large locus to enable V to DJ recombination and diversity of the Igh repertoire. This was a major advance in our understanding of the role of Pax5, the mechanisms underpinning immunoglobulin recombination, and the role and mechanism of loop extrusion more widely in genome organisation.

The current study asks whether this mechanism of loop extrusion also applies in recombination of the Igkappa locus, the second large locus that contributes to antibody formation, which is recombined at the next stage of B cell development, the preB cell. The Igkappa locus structure differs from the Igh structure in the important aspect of having multiple reverse CTCF binding elements in the Igkappa V region, which can participate in loop formation, in contrast to the Igh locus, where the reverse CBEs were confined to the 3' end of the locus.

Overall the data generated is of high quality, using cutting-edge techniques including HiC, Micro-C and VDJ-seq, in ex vivo proB and preB cells the WAPL mutant model, and the RagB1-8 Igh transgene model, which enables assessment of the structure of the intact Igkappa in preB cells, since no V-J recombination has taken place.

The paper provides several interesting findings:

- The authors show elegantly that, in addition to looping across the whole Igkappa from the 3' elements, these internal sites participate in multiple small internal loops, revealing a different multiple loop model for Igkappa conformational change.

- The authors explore the impact of the WAPL promoter mutant described above on Igk recombination in preB cells. Surprisingly, despite increased WAPL expression, there is little effect on Igk V-J recombination. They use HiC to show that this is because there is little change in Igk locus structure. They infer that the Igk locus is insensitive to physiological WAPL expression increases, which they show occur in the transition from proB to preB cells, in a similar fold change to the WAPL promoter mutant, and also insensitive to even higher WAPL increases in WAPL promoter mutant preB cells. Indeed the Igkappa locus tolerates higher levels of WAPL that inhibit

Igh DNA loop extrusion. They further show that the proB-preB transition results in significant changes in genome-wide genome organisation. Thus they show that the two Ig loci undergo conformational change for V(D)J recombination in rather different ways.

My main concern is that one of the messages of the paper is that low levels of WAPL in proB cells enable Igh loop extrusion, while increased levels in preB cells cause loss of looping in the Igh, thereby reversing previously described locus contraction and enabling allelic exclusion. That may very well be the case, and if so would be an extremely exciting advance in our understanding of allelic exclusion, a long-studied but still poorly understood mechanism. The change in WAPL levels, together with the lack of sensitivity of Igkappa to WAPL levels does support that hypothesis, but there are other possible scenarios. WAPL is clearly important, but it is not the only protein whose levels change at the proB to preB transition, so it is possible that changes in other factors also contribute. As the reciprocal complement to this work, knockdown of WAPL in preB cells would be predicted to restore endogenous Igh loop extrusion or allelic exclusion and would definitively answer the question. I am not necessarily suggesting that this experiment needs to be done, but a broader discussion and acknowledgement of alternative scenarios would improve the message of this paper.

Specific comments:

1. The use of the word 'force' in the title seems unnecessary and rather misleading. Which locus is being forced? In physiological conditions of low WAPL expression in proB cells, the Igh employs long-range loop extrusion. In physiological conditions of higher WAPL expression, Igkappa employs long-range looping from Cer-Sis throughout the V region, and additionally is able to generate smaller internal loops.
2. In Figure 1, it is stated that only 3' Vk genes altered/increased with Wapl P1P2, but looks as if Vk15, 19, 13 are also increased.
3. The authors show elegantly by Micro-C and associated analyses that the presence of numerous reverse CBEs in the Igkappa V region allows folding into multiple smaller internal loops, as an alternative to large-scale loop extrusion. This presumably mitigates the impact of cohesin removal due to increased WAPL expression. Page 16: the authors state "The distance shortening induced by the multiple loops likely accounts for the previously observed contraction of the Igk locus in pre-B cells. This folding principle invariably leads to the collision of loops that likely results in the formation of a transient interaction zone". Where is the evidence for the statement 'invariably leads to the collision of loops'? If there is a reference, please provide and discuss. Although 'invariably' implies that the principle can be applied here. Since the Igk locus has many unique conformational aspects, I question whether general principles can be extrapolated without evidence.
4. Additionally, because of the small loops in Ig kappa the authors favour a mechanism of local diffusion with respect to Cer and Sis. It would be helpful to include discussion of previous work on local diffusion and phase separation in Ig loci, in particular Khanna et al Nature Communications 2019.
5. It has not been shown/discussed what changes occur in the Rag B1-8 model ie do the same changes in gene expression/protein expression occur when this model is compared with a Rag knockout model? The authors also have not discussed the issue of whether the wild-type Igh allele behave normally in the Rag B1-8 model. They state that the increased WAPL levels preclude Igh loop extrusion and thus the endogenous Igh allele only recombine the first six V genes. But do they know that/whether the endogenous Ig loci ever engaged in loop extrusion in this model? Arguably early expression of the Igh transgene could impair any opening up of both Ig loci. Again, if the status of the endogenous loci is not known, knockdown of WAPL in the Rag B1-8 model and predicted restoration of Igh loop extrusion would definitively support a role for increased WAPL in its inhibition.
6. Page 21: The authors state: "As half of the VK genes are present in reverse orientation in the Igk locus, convergent RSS alignment by loop extrusion is impossible except for the forward-oriented members of the most 3' proximal VK3 gene family,". The authors have not discussed the

rest of the forward-orientated V κ genes, which are clustered at the 5' end of the V κ region, upstream of E88. I assume they are also theoretically capable of convergent RSS alignment by loop extrusion. Can the authors comment on this – whether it happens, and if not, why not in the context of other loop formation.

The work is original. It draws on previously published data for the purpose of new and legitimate comparison.

The methodology is sound, and the work meets the expected standards in the field. There is enough detail provided in the methods for the work to be reproduced. For some aspects this would be dependent on the availability of the WAPL mouse mutant model generated by the authors.

The work supports the conclusions and claims for the most part.

Overall, this study makes substantial new contributions to our understanding of chromatin conformation mechanisms underpinning I κ recombination and provides key insights into possible mechanisms underpinning allelic exclusion, and advances our understanding of the role of WAPL in genome organisation more widely. Thus in my opinion, pending addressing of the concerns above, it is suitable for a wide range of readership and for Nature Communications.

RESPONSE TO REVIEWERS' COMMENTS

Enclosed, we submit our plan of revision in response to the reviewer's comments (in black). Our point-by-point replies are highlighted in blue.

Reviewer #1 (expertise in VDJ recombination, spatial regulation of antigen receptor loci accessibility):

Recent work by the Busslinger and Alt labs has demonstrated how Pax5- and Wapl-mediated regulation of loop extrusion at the Igh locus accounts for Igh locus contraction and capture of V gene segments for V-to-DJh recombination in pro-B cells. A similar loop extrusion scenario cannot easily explain capture of Vk gene segments for Vk-to-Jk recombination in pre-B cells, since many Igk CBEs and V segments are present in reverse orientation with respect to the recombination center. Here the authors show that unlike the Igh locus in pro-B cells, Igk locus conformation and recombination are relatively insensitive to mutational elevation of Wapl expression in pre-B cells. They attribute relative insensitivity of Igk locus and genome-wide architecture to elevated Wapl protein in pre-B relative to pro-B cells, in a manner that is sufficient to promote architectural changes in Igk, Igh, and genome-wide, with shorter loops in pre-B cells. They also note that low frequency Igk rearrangement in pro-B cells is biased towards distal Vk segments even with Wapl upregulated by mutation, conditions in which rearrangement of distal Vh gene segments is strongly disfavored. Based on these behaviors and maintenance of long distance Igk chromatin contacts under these conditions, they argue for a different mechanism of folding as compared to Igh, whereby Igk loop organization in pre-B cells provokes loop collisions that bring distantly located V segments into contact with the 3' portion of the locus. Finally, they provide evidence that the switch to shorter range looping in pre-B cells can explain Igh allelic exclusion, which affects all but the most proximal Vh genes.

This is a topic of much interest and the results are potentially significant. However, as presented, the work suffers from two major problems (outlined in greater detail below): (1) insufficient quantitative analysis, weakening the conclusions about differences and similarities in rearrangement and genomic contact profiles; and (2) failure to analyze Igk contacts in WT vs Wapl mutant in the absence of recombination; over-reliance on the suboptimal Igk contact profiles in rearranging B cell populations.

We thank the reviewer for the positive evaluation of our manuscript and will comment on the two points mentioned here in the detailed point-by-point reply below.

Specific concerns:

1. There is a lack of clarity on Igk locus contraction. In the introduction (p4) the authors quote ref#35 that shows that Igk contraction initiates in pro-B cells as well as their own earlier work (ref#28) showing it to initiate in pre-B cells. They restate elsewhere (p11) "pro-B ... already undergo contraction of the Igk locus". But later (p15) they seem to relate Igk contraction to the multiple loops that they detect specifically in pre-B cells. Please clarify.

The reviewer is correct that there is a 'confusing' aspect about the citation of our paper (Roldán et al, 2005, ref. 28). In this paper, we discovered for the first time that the *Igk* locus undergoes contraction in *ex vivo* sorted pre-B cells, which is correct and is not disputed in the scientific community. In the same paper, we also analyzed *in vitro* cultured pro-B cells by DNA-FISH analysis, which suggested that the *Igk* locus is not contracted in these pro-B cells. By performing DNA-FISH analysis with *ex vivo* sorted pro-B cells, Stadhouders and Hendriks (2014, ref. 35) later discovered that the *Igk* locus is already contracted in pro-B cells, which is consistent with the fact that *Igk* rearrangements are detected at a low frequency (15%) in pro-B cells (ref. 34). In response of the reviewer's criticism, we have replaced the old sentence (page 4, 2nd paragraph: Contraction of the *Igk* locus is initiated in pro-B cells³⁵ and then maintained in small pre-B cells²⁸) by the following new sentence: Contraction of the *Igk* locus takes place in small pre-B cells²⁸ and was later shown to be initiated already in pro-B cells³⁵.

We did not use the word ‘specifically’ in our submitted manuscript to indicate that the multiple loops at the *Igk* locus are only detected in pre-B cells and by inference not in pro-B cells. We furthermore would like to point out that DNA-FISH analysis allows to determine whether distantly located sequences within the *Igk* locus are present in close proximity, thus indicating *Igk* locus contraction in pro-B and pre-B cells. However, these data cannot explain how *Igk* locus contraction comes about at the mechanistic level. Only our Micro-C analysis of pro-B and pre-B cells was able to demonstrate that the underlying principle of *Igk* locus contraction is the generation of multiple loops in these early B cell types.

2. The authors make comparative statements throughout that are neither quantified nor statistically analyzed. For example, on p7 it is noted that Vk3-4 to 3-12 rearrange more frequently in *Wapl* mutant compared to WT. Can this be presented in a more quantitative way? Similarly, on p12 it is noted that there is a difference in Vk rearrangement between pro- and pre-B cells. Can this be quantified? See also comments about chromatin contacts below.

We thank the reviewer for pointing this out. First, we have performed statistical analysis of the V_K gene rearrangement data based on four or six independent VDJ-seq experiments performed for each pre-B or pro-B cell type, respectively. The V_K gene rearrangement data are shown as mean values with SEM, as described in the figure legends. In response to the reviewer’s comment, we have now added *P*-values to the significantly differentially rearranged V_K genes in Figures 1b,c and 4a as well as in Extended Data Figure 1e. Second, we have systematically analyzed the difference of V_K gene usage across the *Igk* locus, which was determined for each V_K gene by subtracting its mean recombination frequency in *Wapl*^{ΔP1,2/ΔP1,2} pre-B cells from that in *Wapl*^{+/+} pre-B cells (Extended Data Figure 1c). Likewise, we have analyzed the difference of V_K gene usage between *Wapl*^{+/+} and *Wapl*^{ΔP1,2/ΔP1,2} pro-B cells as well as between *Wapl*^{+/+} pro-B and *Wapl*^{+/+} pre-B cells (Extended Data Figure 5b,d). We mention the observed differences of V_K gene usage on pages 7 (top) and 12 in the result section.

3. Fig 2C is confusing. The figure legend indicates that the color scheme represents differential interactions in *Wapl* mutant relative to WT, and this is consistent with the red (+4) and blue (-4) legend in the figure panel. But they should not also code WT as blue and mutant as red, even though blue does represent interactions higher in WT and red, interactions higher in mutant. Pick one approach to define the color scheme.

We apologize for this apparent confusion. We have improved the labeling (*Wapl*^{ΔP1,2/ΔP1,2} pre-B / *Wapl*^{+/+} pre-B cells) to indicate that this ratio was plotted in Figure 2c. The colors of the indicated genotypes below the contact matrix in Figure 2c are correct for the following reasons. The loops in the TAD range are increased in *Wapl*^{+/+} pre-B cells compared with *Wapl*^{ΔP1,2/ΔP1,2} pre-B cells, as shown in Figure 2a. Hence, the ratio of *Wapl*^{ΔP1,2/ΔP1,2} pre-B / *Wapl*^{+/+} pre-B cells is negative (blue), supporting the blue color for the *Wapl*^{+/+} genotype. In contrast, the compartmentalization is increased in *Wapl*^{ΔP1,2/ΔP1,2} pre-B cells relative to *Wapl*^{+/+} pre-B cells (Figure 2a) and hence the ratio of *Wapl*^{ΔP1,2/ΔP1,2} pre-B / *Wapl*^{+/+} pre-B cells is positive (red), thus supporting the assignment of the red color to the *Wapl*^{ΔP1,2/ΔP1,2} genotype. We now explain the color scheme in the legend of Figure 2c and the new Extended Data Figure 3f.

4. Can the authors further elaborate on the nature of the changes in the contact maps (Fig 2b,c)? It looks as if the regions showing reduced interactions in mutant represent contacts between compartments A and B (white in b becomes blue in c). Is this interpretation correct, and if so, does it mean that elevated *Wapl* modestly increases insulation between nearby opposing compartments? Reciprocally, in Fig 3c, is lower *Wapl* in pro-B is associated with slight breakdown of compartmentalization.

The interpretation of the data by the reviewer is correct. Lower *Wapl* expression causes a breakdown of compartmentalization as longer loops interfere with the compartment structure (Wutz et al., 2017, EMBO J.36, 3573-3599; Haarhuis et al., 2017, Cell 169,693-707). We have

shown this also for pro-B cells in Figure 5c of Hill et al. (2020, Nature 584, 142-147). We mention this fact on page 10 (bottom) in the context of $Wapl^{low}$ pro-B cells and $Wapl^{high}$ pre-B cells.

5. The authors argue that changes in chromatin organization between pro-B and pre-B are due to a 2.2x increase in $Wapl$ expression. I believe they only present one piece of data that moves this beyond a correlation, namely, that $Wapl$ mutant pro-B cells display a distribution of contact frequencies that is much more similar to that of WT pre-B than WT pro-B (compare Fig 3f to Fig 3b). Given the importance of this result, it would be good to see a bit more primary data, for example the Hi-C contact matrix to compare to that in Fig 3c).

In response to the reviewer's request, we now show more 'primary' data in the new Extended Data Figure 4, which unequivocally indicates that the chromosomal architecture is similar in $Wapl^{\Delta P1,2/\Delta P1,2}$ pro-B cells and $Wapl^{+/+}$ pre-B cells.

6. The description of differences in Igk contact maps for WT pre-B (Fig 1d), WT pro-B and mutant pro-B (Fig 4b) is vague and non-quantitative. The same for the comparison of Igh to Igk in mutant pro-B (Extended 3c). In particular, the maintenance of long-distance Igk contacts in mutant pro-B cells, and a clear difference between Igh and Igk , are not as obvious as one might think from reading the text. And it is hard to understand how it is possible to meaningfully compare Igk contact maps between pro-B and pre-B since the extent of rearrangement is so different in the two compartments.

As requested by the reviewer, we have now devoted the new Extended Data Figure 2 to the bioinformatic quantification of the Hi-C data at the Igk locus in $Wapl^{+/+}$, $Wapl^{\Delta P1,2/\Delta P1,2}$ and $Igh^{B1-8hi/+} Rag2^{-/-}$ pre-B cells (panel a-c; corresponding to Figure 1d) and in $Wapl^{+/+}$ and $Wapl^{\Delta P1,2/\Delta P1,2}$ pro-B cells (panel d-f; corresponding to Figure 4b). These new data clearly demonstrate that the internal architecture of the Igk locus and the contact frequencies along the stripe emanating from the Igk 3' end are similar in $Wapl^{+/+}$, $Wapl^{\Delta P1,2/\Delta P1,2}$ and $Igh^{B1-8hi/+} Rag2^{-/-}$ pre-B cells (a-c), as mentioned on page 7 (bottom). In contrast, the Igk architecture differs between $Wapl^{+/+}$ and $Wapl^{\Delta P1,2/\Delta P1,2}$ pro-B cells (panel d-f), as mentioned on page 13 (top).

However, we do not understand why and how we could quantify the contact matrices of the two different Igh and Igk loci (Extended Data Figure 5e [old 3c]) that use distinct folding principles and consist of different sequences.

7. (1) The micro-C analysis of Igk interactions on a R2KO background provides a much cleaner and more highly resolved look at interactions across the Igk locus. (2) The authors should have analyzed $Wapl$ -mutant mice on this background, thereby simplifying the manuscript and strengthening the conclusions.

(1) Our conclusions based on Hi-C data were confirmed by the Micro-C data, wherever we could analyze data generated with both methods for the same cell type. Hence, there is no reason to believe that replacing the Hi-C data by Micro-C data in Figure 1d, 2, 3 and 4 would lead to different conclusion. In this context, it is also important to note that the Micro-C method was only published in 2020 (ref. 45 and 46). On the other hand, we performed the Hi-C analyses before this time. Both the Micro-C and Hi-C methods require sequencing at a high depth of 500 – 1'000 million sequence reads, which makes these experiments extremely expensive. It is therefore impossible for us to replace the Hi-C data with Micro-C data for financial and time reasons.

(2) The reviewer asks us to analyze pre-B cells of $Wapl^{\Delta P1,2/\Delta P1,2} Igh^{B1-8hi/+} Rag2^{-/-}$ mice that are difficult to generate due to their complex genotype. The result would not be worth the breeding effort, as we have already shown that the contact matrices of the Igk locus in $Wapl^{+/+} Rag2^{+/+}$ and $Wapl^{\Delta P1,2/\Delta P1,2} Rag2^{+/+}$ pre-B cells are very similar compared with those of $Igh^{B1-8hi/+} Rag2^{-/-}$ pre-B cells (Figure 1d and new Extended Data Figure 2a-c). Moreover, the generation of $Wapl^{\Delta P1,2/\Delta P1,2} Igh^{B1-8hi/+} Rag2^{-/-}$ mice would take about a year and is therefore beyond the scope of this manuscript.

8. It was noted in Fig 4A that Igk rearrangements in pro-B are weighted more distally than in pre-B. Can the authors relate quantitative differences in long-distance interactions in the micro-C

experiments to this regional difference in V_K rearrangement? With this in mind, it looks as if there is a more well-defined loop organization in the distal as compared to the proximal half of the V_K region in pro-B cells. This difference is not as apparent in pre-B cells, where both distal and proximal regions are structured.

As suggested by the reviewer, we have quantified the contact frequencies along the Micro-C stripe emanating from the 3' end of the *Igk* locus in *Rag2*^{-/-} pro-B and *Igh*^{B1-8hi/+} *Rag2*^{-/-} pre-B (new Extended Data Figure 6a). This analysis revealed that the contact frequencies in the distal *Igk* region were higher in *Rag2*^{-/-} pro-B cells compared with *Igh*^{B1-8hi/+} *Rag2*^{-/-} pre-B cells. Moreover, correlation of the long-distance interactions with the differential V_K gene usage indicated that several distal V_K genes in the distal *Igk* region were preferentially recombined in *Rag2*^{+/+} pro-B cells relative to *Rag2*^{+/+} pre-B cells (new Extended Data Figure 6b).

9. It is suggested (p14) that collisions at the bases of different loops creates the continuum of interactions that form the stripe emanating from Cer. It is not explicitly stated, but when the authors refer to constant turnover and formation of new loops, do they include as well the process of loop extrusion within these shorter loops as contributing to this continuum of contacts?

Yes, this is exactly what our video explains.

10. Extended data Fig 5c,d presents Hi-C comparison of *Igh* in WT pro-B to *Rag2*KO pre-B. What is the point of this comparison, which is far from ideal since pro-B but not pre-B will have undergone rearrangement? And the more valid comparison, of *Rag2*KO pro-B to *Rag2*KO pre-B, is presented in the panels immediately above. Justify or remove.

Our intention for showing the data of Extended Data Figure 7a-d was to demonstrate that the Micro-C data (top row; panel a and b) and Hi-C data (bottom row; panel c and d) both reveal the same architectural features at the *Igh* locus in pro-B and pre-B cells, which is clearly supported by the data shown. The argument that all wild-type pro-B cells have undergone V_H-DJ_H recombination is incorrect. Pro-B cells are in the process of undergoing V_H-DJ_H recombination. Upon successful rearrangement, the pro-B cells rapidly differentiate to large pre-B cells, and hence functional VDJ_H-rearranged alleles do not accumulate in the pro-B cell population. It is therefore justified to show the Hi-C data of *Wapl*^{+/+} pro-B cells. Furthermore, we would like to point out that our detailed bioinformatic comparison of the chromosomal architecture at the *Igk* locus did not reveal significant differences between RAG-sufficient *Wapl*^{+/+} pre-B cells and RAG-deficient *Igh*^{B1-8hi/+} *Rag2*^{-/-} pre-B cells (Extended Data Figure 2a-c).

11. (1) The authors propose that *Igk* contraction is a function of its multiple loop organization (p18). Can they relate this to the *Igk* loop organization in pro-B, in which the locus is contracted (see above) but in which loops are somewhat less well defined? (2) Can they contrast this with *Igh*, which also shows internal looping in pre-B (Fig 5b), where the locus is decontracted?

(1) The first part of the question has been largely answered in response to point 8 (see above). By bioinformatic quantification, we could demonstrate that the long-distance interactions in the distal *Igk* region are increased in *Rag2*^{-/-} pro-B cells compared with *Igh*^{B1-8hi/+} *Rag2*^{-/-} pre-B cells (Extended Data Figure 6a). This furthermore suggested that more internal loops are formed in the distal *Igk* region in *Rag2*^{-/-} pro-B cells, which could explain the preferential usage of distal V_K genes for recombination in wild-type pro-B cells relative to wild-type pre-B cells (Extended Data Figure 6b).

(2) The suggested comparison of internal loops between the *Igh* and *Igk* loci may be based on a misunderstanding. The internal loops at the *Igk* locus are generated due to the presence of convergent CTCF-binding sites. In contrast, all CTCF-binding sites in the V_H gene cluster of the *Igh* locus are present in the same forward orientation, and hence the internal structures in the *Igh* locus cannot be generated by loops that are arrested at convergent CTCF-binding sites. Given this difference between the *Igh* and *Igk* loci, we do not see how we can compare the internal structures at the *Igk* locus in pre-B cells with those at the *Igh* locus in pre-B cells.

Reviewer #2 (expertise in 3D genome conformation, hi-C, micro-C):

This is an important piece of work that addresses a gap in the field of 3D organization and recombination of light chain loci in B cells. As stated, the role of cohesin and loop extrusion is shown clearly for recombination at Igh and T cell receptor loci but is inconsistent with Igk. The reasonable model the authors propose here for Igk is its contraction by many internal loops facilitating RAG-mediated cleavage and recombination. I am very much in favor of seeing this paper published although I must say it is extremely hard to follow at times (at least for this reviewer). I have some comments below, which are important to address.

We thank the reviewer for a very positive evaluation of our manuscript.

- The authors need to perform a proper analysis of loops/contacts that are differential between the compared conditions. This is important throughout the paper but more specifically for Figure 2d and 3d and how/if those observed patterns generalize genome-wide.

We thank the reviewer for this interesting suggestion. We have reanalyzed the respective Hi-C data to determine the loop lengths and loop numbers in a systematic and genome-wide manner. These data are now shown for the following three comparisons: 1) *Wapl*^{+/+} versus *Wapl*^{ΔP1,2/ΔP1,2} pre-B cells in Figure 2e and Extended Data Figure 3a,b, which revealed similar loop lengths and numbers in the two cell types; 2) *Wapl*^{+/+} pro-B versus *Wapl*^{+/+} pre-B cells in Figure 3e and Extended Data Figure 3g, which indicated strong differences in the loop lengths and numbers between these cell types; 3) *Wapl*^{+/+} pre-B versus *Wapl*^{ΔP1,2/ΔP1,2} pro-B cells in Figure 3e and Extended Data Figure 4d, which uncovered comparable loop lengths and similar loop numbers in these cell types.

- (1) The sentence that says "as a 2.2-fold increase of *Wapl* protein expression from *Wapl*^{+/+} pro-B to *Wapl*^{+/+} pre-B cells ..." missed the point that such increase PLUS the cell state difference is what causes drastic changes in comparison to within pre-B cells comparison with differences in *Wapl* levels. This statement has to be corrected to accurately reflect that. They are later careful and add "likely" to the last sentence of this section. (2) Related to this, the comparison of *Wapl* del(P1,2) pro-B cells to *Wapl* WT+ pre-B cells should be done more thoroughly beyond showing Figure 3f if the claim is that the two are very similar.

(1) Formally, it could be argued that the developmental stage also contributes to the change in chromosomal architecture in pre-B cells. However, we have documented before that mature B cells and *Pax5*^{Δ/Δ} progenitors have a similar architecture (Hill et al., 2020, ref. 20), as we show in Figure 1 (for reviewer). The newly generated Hi-C data of pre-B cells are also

Figure 1 (for reviewer). Hi-C contact matrices of chromosome 12 based on published Hi-C data (Hill et al., 2020, Nature, ref. 20) except for wild-type (WT) pre-B cells (this study).

very similar to those of *Pax5*^{Δ/Δ} progenitors, WT pre-B cells and WT mature B cells, but strongly differ from WT pro-B cells. Hence, the pro-B cells are the exception with regard to the chromosomal architecture, while the pre-B cells resemble other B cell types (argument 1). Furthermore, our new bioinformatic analysis (Extended Data Figure 4) has conclusively demonstrated that the chromosomal architecture is similar in *Wapl*^{ΔP1,2/ΔP1,2} pro-B cells and

Wapl^{+/+} pre-B cells, indicating that similar *Wapl* expression at different developmental stages is responsible for similar architectural changes (argument 2). Moreover, as pointed out in the manuscript on page 19, 'the observed 2.2-fold increase of *Wapl* protein expression in pre-B cells, possibly due to enhanced protein translation or stabilization, likely causes these global changes in chromosomal architecture, consistent with our previous finding that a 1.7- to 1.9-fold increase of *Wapl* expression in *Wapl*^{ΔP1/ΔP1} and *Wapl*^{ΔP1,2/+} pro-B cells abolishes V_H gene recombination across the *Igh* locus due to drastic architectural changes' (Hill et al., 2020, ref. 20). In other words, we have already shown that a 1.7- to 1.9-fold increase of *Wapl* expression is sufficient to induce the drastic changes that we have described here for the transition from wild-type pro-B to wild-type pre-B cells (argument 3). Based on these strong arguments, there is no good reason why we should tone our *Wapl* statement down in the manuscript. We have, however, inserted the word 'likely' (underlined above) in the revised manuscript version on page 19.

(2) In response to the reviewer's request, we now show more 'primary' data in the new Extended Data Figure 4, which unequivocally indicates that the chromosomal architecture is similar in *Wapl*^{ΔP1,2/ΔP1,2} pro-B cells and *Wapl*^{+/+} pre-B cells.

- The conclusion that "increased *Wapl* protein expression had a smaller effect on both differential gene expression and genomic architecture in pre-B cells relative to pro-B cells" has to be supported by uniform analysis of RNA-seq data between the previous Pax5-*Wapl* paper and this paper. It should be the same analysis rather than a similar analysis. If that is already the case, the authors should state so.

The reviewer is right about the importance of a uniform analysis of the RNA-seq data. The *Wapl*^{ΔP1,2/ΔP1,2} and *Wapl*^{+/+} pre-B cells were sequenced at the same time as the *Wapl*^{ΔP1,2/ΔP1,2} and *Wapl*^{+/+} pro-B cells (Supplementary Table 3, experiment numbers 72347-72354, published in Hill et al., 2020, ref. 20). As these RNA-seq experiments were performed at the same time, there are no batch effects to be considered, and therefore it should be justified to determine the differentially expressed genes in both pro-B and pre-B cells. We now mention in the legend of Extended Data Figure 3d that both RNA-seq experiments were performed at the same time. Importantly, also the bioinformatic analysis of these RNA-seq data was performed in the same way for all 4 cell types.

- The lack of translation between mRNA level differences and protein-level measurements for *Wapl* between pro- and pre-B cells is certainly interesting and warrants further investigation. However, it is unclear to me how higher *Wapl* in pre-B cells may lead to longer-range interactions. This seems mainly due to normalization of the scaling curves such that pro-B cells having more <5Mb interactions translate into them having relatively less that are >5Mb. A difference map, such as the one in Fig 2c is needed to strengthen this argument visually.

As suggested by the reviewer, we now show a differential Hi-C contact matrix of chromosome 1 for comparing the chromosomal architecture of *Wapl*^{+/+} pro-B and *Wapl*^{+/+} pre-B cells in the new Extended Data Figure 3f. These new data confirm that there is a dramatic difference of the chromosomal architecture between *Wapl*^{+/+} pro-B and *Wapl*^{+/+} pre-B cells. This strong change is consistent with previous reports that lower *Wapl* expression causes a breakdown of compartmentalization as longer loops interfere with the compartment structure (Wutz et al., 2017, EMBO J.36, 3573-3599; Haarhuis et al., 2017, Cell 169,693-707). We have previously documented this also for pro-B cells as shown in Figure 5c of Hill et al. (2020, Nature, ref. 20). Consequently, higher *Wapl* expression leads to increased compartmentalization as there are fewer longer loops that could interfere with this process as shown for *Wapl*^{+/+} pre-B cells.

- The interpretation of Figure 4b "the structures of the sub-TADs were, however, less well defined in *Wapl*^{+/+}" has to be accompanied by quantification. The heatmaps are not strikingly different.

In response to the reviewer's comment, we have performed a detailed bioinformatic quantification of the Hi-C data at the *Igk* locus in *Wapl*^{+/+} and *Wapl*^{ΔP1,2/ΔP1,2} pro-B cells, which is now shown in the Extended Data Figure 2d-f. These new data clearly demonstrate that the internal architecture of the *Igk* locus and the contact frequencies along the stripe emanating from

the *Igk* 3' end are different between *Wapl*^{+/+} and *Wapl*^{ΔP1,2/ΔP1,2} pro-B cells, as mentioned on page 13 (top) in the result section.

Minor

- contract frequencies -> contact frequencies

We corrected this mistake throughout the manuscript.

- *Wapl*^{high} terminology is used sporadically. I suggest they early on define *Wapl* del(P1,2) as *Wapl*^{high} and go with it. It makes it much easier to read.

The terms *Wapl*^{high} and *Wapl*^{low} refer to a relative difference in *Wapl* expression. In this generic sense, these terms cannot be used to replace the relevant genotypes, as we discuss *Wapl* protein differences between *Wapl*^{low} *Wapl*^{+/+} pre-B and *Wapl*^{high} *Wapl*^{ΔP1,2/ΔP1,2} pre-B cells, between *Wapl*^{low} *Wapl*^{+/+} pro-B and *Wapl*^{high} *Wapl*^{+/+} pre-B cells as well as between *Wapl*^{low} *Wapl*^{+/+} pro-B and *Wapl*^{high} *Wapl*^{ΔP1,2/ΔP1,2} pro-B cells. For this reason, we could not broadly apply the suggested terms throughout the manuscript.

- The video clip is a nice attempt to summarize their model but it is far from self-explanatory. Not sure if this can be made better by more text and/or narration, the reference to the video in the text has a good description.

We thank the reviewer for pointing this out. We have added an explanatory slide at the beginning of the video, as suggested by the reviewer.

Reviewer #3 (expertise in Ig loci regulation, VDJ recombination, chromatin accessibility during VDJ recombination):

This is a very interesting study that follows up on the authors' recent seminal study (Hill et al 2020) which showed that the B cell specific transcription factor, Pax5 orchestrates its widespread effects on gene regulation in progenitor B cells via a newly discovered mechanism. Specifically, it binds and represses the promoter of the WAPL cohesin release factor gene, thereby reducing WAPL expression in proB cells. That paper shows that this leads to reduced cohesin residence time on the large immunoglobulin heavy chain locus, thereby inhibiting DNA loop extrusion, shown by the authors and others to provide the underlying mechanism that brings V and D genes together in this very large locus to enable V to DJ recombination and diversity of the Igh repertoire. This was a major advance in our understanding of the role of Pax5, the mechanisms underpinning immunoglobulin recombination, and the role and mechanism of loop extrusion more widely in genome organisation.

The current study asks whether this mechanism of loop extrusion also applies in recombination of the Igkappa locus, the second large locus that contributes to antibody formation, which is recombined at the next stage of B cell development, the preB cell. The Igkappa locus structure differs from the Igh structure in the important aspect of having multiple reverse CTCF binding elements in the Igkappa V region, which can participate in loop formation, in contrast to the Igh locus, where the reverse CBEs were confined to the 3' end of the locus.

Overall the data generated is of high quality, using cutting-edge techniques including HiC, Micro-C and VDJ-seq, in ex vivo proB and preB cells the WAPL mutant model, and the RagB1-8 Igh transgene model, which enables assessment of the structure of the intact Igkappa in preB cells, since no V-J recombination has taken place.

The paper provides several interesting findings:

- The authors show elegantly that, in addition to looping across the whole Igkappa from the 3' elements, these internal sites participate in multiple small internal loops, revealing a different multiple loop model for Igkappa conformational change.

- The authors explore the impact of the WAPL promoter mutant described above on Igk recombination in preB cells. Surprisingly, despite increased WAPL expression, there is little effect on Igk V-J recombination. They use HiC to show that this is because there is little change in Igk locus structure. They infer that the Igk locus is insensitive to physiological WAPL expression increases, which they show occur in the transition from proB to preB cells, in a similar fold change to the WAPL promoter mutant, and also insensitive to even higher WAPL increases in WAPL promoter mutant preB cells. Indeed the Igkappa locus tolerates higher levels of WAPL that inhibit Igh DNA loop extrusion. They further show that the proB-preB transition results in significant changes in genome-wide genome organisation. Thus they show that the two Ig loci undergo conformational change for V(D)J recombination in rather different ways.

We thank the reviewer for the very positive evaluation of our manuscript.

My main concern is that one of the messages of the paper is that low levels of WAPL in proB cells enable Igh loop extrusion, while increased levels in preB cells cause loss of looping in the Igh, thereby reversing previously described locus contraction and enabling allelic exclusion. That may very well be the case, and if so would be an extremely exciting advance in our understanding of allelic exclusion, a long-studied but still poorly understood mechanism. The change in WAPL levels, together with the lack of sensitivity of Igkappa to WAPL levels does support that hypothesis, but there are other possible scenarios. WAPL is clearly important, but it is not the only protein whose levels change at the proB to preB transition, so it is possible that changes in other factors also contribute. As the reciprocal complement to this work, knockdown of WAPL in preB cells would be predicted to restore endogenous Igh loop extrusion or allelic exclusion and would definitively answer the question. I am not necessarily suggesting that this experiment needs to be done, but a broader discussion and acknowledgement of alternative scenarios would improve the message of this paper.

The argument of the reviewer is valid. We will qualify our statement that the increased *Wapl* expression is responsible for allelic exclusion of the second, non-rearranged *Igh* allele in pre-B cells. In the absence of a genetic experiment demonstrating that reduced *Wapl* expression in pre-B cells causes allelic exclusion, we have toned the statements dealing with *Igh* allelic exclusion down in the abstract (page 2), introduction (page 5), result section (page 18) and discussion (page 22).

As mentioned in our response to reviewer #2 on page 6 (top) of this point-by-point reply and also mentioned in the manuscript on page 19, 'the observed 2.2-fold increase of *Wapl* protein expression in pre-B cells, possibly due to enhanced protein translation or stabilization, likely causes these global changes in chromosomal architecture, consistent with our previous finding that a 1.7- to 1.9-fold increase of *Wapl* expression in *Wapl*^{ΔP1/ΔP1} and *Wapl*^{ΔP1.2/+} pro-B cells abolishes V_H gene recombination across the *Igh* locus due to drastic architectural changes' (Hill et al., 2020, ref. 20). In other words, we have already shown that a 1.7- to 1.9-fold increase of *Wapl* expression is sufficient to induce the drastic architectural changes that we have described here for the transition from wild-type pro-B to wild-type pre-B cells. Based on this argument, there is no good reason why we should tone our *Wapl* statement down in the manuscript by invoking other critical proteins in the change of the chromosomal architecture in pre-B cells. However, in response to this reviewer's comment, we have inserted the word 'likely' (underlined above) in the revised manuscript version on page 19.

Unfortunately, it has so far been impossible to propagate *ex vivo* sorted pre-B cells in an *in vitro* culture system, unless they are immortalized by the v-Abl virus, which creates a very artificial situation. For this reason, it is technically not possible to knockdown *Wapl* expression in *ex vivo* sorted pre-B cells. Moreover, the reviewer did not insist on performing this experiment. Furthermore, conditional *Wapl* mutagenesis is also not a feasible option, as the absence of *Wapl* causes loss of *Cd79a-Cre Wapl*^{fl/fl} pro-B cells and all subsequent B cell stages, as previous shown (Hill et al., 2020, ref. 20). In order to settle the allelic exclusion issue, we have just recently obtained a *Wapl*^{Aid/+} founder mouse, which should in the long run facilitate acute degradation of the *Wapl* protein *in vivo* through its C-terminally added auxin-inducible degron (*Aid*) sequence in

pre-B cells. However, this experiment is beyond the scope of this manuscript, as it will take us at least a year to generate experimental *Igh*^{B1-8hi/+} *Rosa26*^{Tir1-F74G/+} *Wapl*^{Aid/Aid} mice, in case the *Wapl*^{Aid} allele will prove to be functional after all.

Specific comments:

1. The use of the word 'force' in the title seems unnecessary and rather misleading. Which locus is being forced? In physiological conditions of low WAPL expression in proB cells, the Igh employs long-range loop extrusion. In physiological conditions of higher WAPL expression, Igkappa employs long-range looping from Cer-Sis throughout the V region, and additionally is able to generate smaller internal loops.

We have changed the title of the manuscript by eliminating the word "forced", as suggested by the reviewer.

2. In Figure 1, it is stated that only 3' V_K genes altered/increased with Wapl P1P2, but looks as if Vk15, 19, 13 are also increased.

We thank the reviewer for pointing this out. In response to this reviewer and reviewer #1 (point 2). we have systematically analyzed the difference of V_K gene usage across the *Igk* locus, which was determined for each V_K gene by subtracting its mean recombination frequency in *Wapl*^{ΔP1,2/ΔP1,2} pre-B cells from that in *Wapl*^{+/+} pre-B cells (Extended Data Figure 1c). Likewise, we have analyzed the difference of V_K gene usage between *Wapl*^{+/+} and *Wapl*^{ΔP1,2/ΔP1,2} pro-B cells as well as between *Wapl*^{+/+} pro-B and *Wapl*^{+/+} pre-B cells (Extended Data Figure 5b,d). We mention the observed differences of V_K gene usage on pages 7 (top) and 12 in the result section.

3. The authors show elegantly by Micro-C and associated analyses that the presence of numerous reverse CBEs in the Igkappa V region allows folding into multiple smaller internal loops, as an alternative to large-scale loop extrusion. This presumably mitigates the impact of cohesin removal due to increased WAPL expression. Page 16: the authors state "The distance shortening induced by the multiple loops likely accounts for the previously observed contraction of the Igk locus in pre-B cells. This folding principle invariably leads to the collision of loops that likely results in the formation of a transient interaction zone". Where is the evidence for the statement 'invariably leads to the collision of loops'? If there is a reference, please provide and discuss. Although 'invariably' implies that the principle can be applied here. Since the Igk locus has many unique conformational aspects, I question whether general principles can be extrapolated without evidence.

It is common knowledge in the cohesin field that loops, which extend by loop extrusion, run into each other but cannot jump over each other, thus leading to collision of the loops as discussed by Fudenberg et al. (2016, Cell Rep., ref 14) and Costantino et al. (2020, eLife, ref. 47). We now mention these citations in the result section (page 15, bottom) and discussion (page 20) of the revised manuscript.

4. Additionally, because of the small loops in Ig kappa the authors favour a mechanism of local diffusion with respect to Cer and Sis. It would be helpful to include discussion of previous work on local diffusion and phase separation in Ig loci, in particular Khanna et al Nature Communications 2019.

As suggested by the reviewer, we now cite Khanna et al. (2019, Nat. Commun., ref. 62) on pages 22 (top) and 36 for the local diffusion that is required for the proper alignment of convergent *Igk* RSS elements prior to RAG-mediated cleavage.

5. (1) It has not been shown/discussed what changes occur in the Rag B1-8 model ie do the same changes in gene expression/protein expression occur when this model is compared with a Rag knockout model? (2) The authors also have not discussed the issue of whether the wild-type Igh allele behave normally in the Rag B1-8 model. They state that the increased WAPL levels preclude Igh loop extrusion and thus the endogenous Igh allele only recombine the first six V

genes. But do they know that/whether the endogenous Ig loci ever engaged in loop extrusion in this model? Arguably early expression of the Igh transgene could impair any opening up of both Ig loci. (3) Again, if the status of the endogenous loci is not known, knockdown of WAPL in the Rag B1-8 model and predicted restoration of Igh loop extrusion would definitively support a role for increased WAPL in its inhibition.

(1) The statement of the reviewer is not clearly formulated but suggests that we should compare the gene expression of *Igh*^{B1-8hi/+} *Rag2*^{-/-} pre-B cells with that of *Rag2*^{-/-} pre-B cells. However, *Rag2*^{-/-} pre-B cells cannot be generated in the absence of a functionally rearranged Ig μ transgene. Hence, the suggested comparison cannot be done.

(2) First, it is important to mention that every gene engages in loop extrusion. The question here is whether the second wild-type *Igh* allele engages in extended loop extrusion during a potentially brief transit through the pro-B cell stage. The answer must be no, as otherwise more than the six most 3' proximal V_H genes should be present in a recombined state in *Igh*^{B1-8hi/+} *Rag2*^{+/+} pre-B cells. The next question to answer is whether the six most 3' proximal V_H genes may recombine already in the uncommitted lymphoid progenitors undergoing D_H-J_H recombination. However, this is clearly ruled out by the paper of Guo et al. (2011, Nature 477, 424-430), demonstrating that the insulator function of the IGCR1 element (located in the region between the V_H and D_H elements) prevents proximal V_H-D_H recombination in early lymphoid progenitors. Consequently, the six most 3' proximal V_H genes of the second wild-type *Igh* allele must have been recombined at the pre-B cells stage in *Igh*^{B1-8hi/+} *Rag2*^{+/+} mice. Moreover, it is important to mention that Ig μ and TCR β transgenic mice are well-accepted models for studying allelic exclusion at the *Igh* or *Tcrb* locus in B and T cells, respectively.

(3) As explained above under "main concern", it is not possible to culture *ex vivo* sorted pre-B cells to perform the suggested Wapl knockdown experiment. Moreover, the reviewer did also not insist on performing this experiment under "main concern".

6. Page 21: The authors state: "As half of the VK genes are present in reverse orientation in the Iglk locus, convergent RSS alignment by loop extrusion is impossible except for the forward-oriented members of the most 3' proximal VK3 gene family,". The authors have not discussed the rest of the forward-orientated Vk genes, which are clustered at the 5' end of the V κ region, upstream of E88. I assume they are also theoretically capable of convergent RSS alignment by loop extrusion. Can the authors comment on this – whether it happens, and if not, why not in the context of other loop formation.

Most of the forward-oriented V κ genes together with the interspersed forward-oriented CTCF-binding element (CBEs) are present in the distal region of the *Iglk* locus. These forward-oriented V κ genes are, however, separated from the J κ elements at the *Iglk* 3' end by a long stretch of reverse-oriented CBE's, which promote the formation of internal loops in the V κ gene cluster. These internal loops prevent, however, the convergent alignment of the distal forward-oriented V κ genes with the J κ elements by extended loop extrusion, which by the way cannot take place due to the high Wapl expression in pre-B cells.

The work is original. it draws on previously published data for the purpose of new and legitimate comparison.

The methodology is sound, and the work meets the expected standards in the field. There is enough detail provided in the methods for the work to be reproduced. For some aspects this would be dependent on the availability of the WAPL mouse mutant model generated by the authors.

The work supports the conclusions and claims for the most part.

Overall, this study makes substantial new contributions to our understanding of chromatin conformation mechanisms underpinning Iglkappa recombination and provides key insights into possible mechanisms underpinning allelic exclusion, and advances our understanding of the role

of WAPL in genome organisation more widely. Thus in my opinion, pending addressing of the concerns above, it is suitable for a wide range of readership and for Nature Communications.

We thank the reviewer once more for the very positive evaluation of our manuscript.

REVIEWER COMMENTS

Reviewer #1 (expert in VDJ recombination, spatial regulation of antigen receptor loci accessibility):

The authors have done a good job of responding to the review. Congratulations on an outstanding piece of work which represents a highly important contribution to the field.

Reviewer #2 (expert in 3D genome conformation, hi-C, micro-C):

The authors have done a good job of addressing many of the concerns. I have some remaining concerns (outlined below) about their interpretation of the results and the numbers when it comes to comparing conditions.

1. A systematic comparative analysis between the compared conditions is critical for most conclusions they are making. Although this is now improved, I do not see any methodology, beyond saying the samples were downsampled to the same depth, about how the numbers in Extended Figure 3b, 3g and similar plots were extracted. It is not an exact overlap as that would require the common number to be identical. What type of overlap has been used here? Is there any other filtering for calling a loop common or unique?

2. For Figure 1 (for the reviewer), the conclusions about similarities and differences that are mentioned are from visual inspection and are not necessarily always obvious. Heatmap scales can easily make visualizations look more similar to or different from each other. Also, while they say Extended Figure 4 "unequivocally" shows the similarity, the loop (number) panel in this figure shows half the loops are different between the two conditions - not too far from the same analysis in Extended Figure 3g comparing *Wapl*^{+/+} pre-B cells to *Wapl*^{+/+} pro-B cells, a comparison referred to as having strong differences. I understand this is partially due to loop callers and their issues in reproducibly reporting the same pixel as loops. At this point, all I can suggest is the authors clearly indicate where their conclusions are based on visual inspection instead of systematic comparisons or objective interpretation of the loop numbers they present as "our visual inspection of heatmaps suggested that this etc".

Reviewer #3 (expert in Ig loci regulation, VDJ recombination, chromatin accessibility during VDJ recombination):

I consider that the authors have addressed my concerns in their detailed rebuttal letter to myself and the other two reviewers. I now recommend this manuscript for publication in Nature Communications.

Anne Corcoran Reviewer 3

RESPONSE TO REVIEWERS' COMMENTS

Enclosed, we submit our response to the points raised by the reviewers (in black). Our replies are highlighted in blue and the new text in the manuscript and Method section are indicated in red. ithe revised Method section.

Reviewer #1 (expert in VDJ recombination, spatial regulation of antigen receptor loci accessibility):

The authors have done a good job of responding to the review. Congratulations on an outstanding piece of work which represents a highly important contribution to the field.

We are very grateful to the reviewer for acknowledging that we did good job in revising our manuscript. Many thanks for the congratulations.

Reviewer #3 (expert in Ig loci regulation, VDJ recombination, chromatin accessibility during VDJ recombination):

I consider that the authors have addressed my concerns in their detailed rebuttal letter to myself and the other two reviewers. I now recommend this manuscript for publication in Nature Communications.

We thank the reviewer for recommending our paper for publication.

Reviewer #2 (expert in 3D genome conformation, hi-C, micro-C):

The authors have done a good job of addressing many of the concerns. I have some remaining concerns (outlined below) about their interpretation of the results and the numbers when it comes to comparing conditions.

1. A systematic comparative analysis between the compared conditions is critical for most conclusions they are making. Although this is now improved, I do not see any methodology, beyond saying the samples were downsampled to the same depth, about how the numbers in Extended Figure 3b, 3g and similar plots were extracted. It is not an exact overlap as that would require the common number to be identical. What type of overlap has been used here? Is there any other filtering for calling a loop common or unique?

For our previous analyses, we used a loop anchor-based method for comparing chromatin loops between two cell types, which resulted in complex patterns of loops and ambiguous loop clusters. This may have been the reason why the common loop numbers were different between the two cell types compared. We have now used a simpler method for loop calling, which is described in great detail on page 7 of the Methods section. The numbers of common loops are now almost identical for the comparison of *Wapl*^{+/+} and *Wapl*^{ΔP1,2/ΔP1,2} pre-B cells (3,027 vs 3,033; Extended Data Fig. 3b) as well as *Wapl*^{+/+} pre-B and *Wapl*^{ΔP1,2/ΔP1,2} pro-B cells (2,505 vs 2,507; Extended Data Fig. 4d) and are similar for the comparison of *Wapl*^{+/+} pro-B and *Wapl*^{+/+} pre-B cells (3,219 vs 3,091; Extended Data Fig. 3g).

2. **A)** For Figure 1 (for the reviewer), the conclusions about similarities and differences that are mentioned are from visual inspection and are not necessarily always obvious. Heatmap scales can easily make visualizations look more similar to or different from each other. **B)** Also, while they say Extended Figure 4 "unequivocally" shows the similarity, the loop (number) panel in this figure shows half the loops are different between the two conditions - not too far from the same analysis in Extended Figure 3g comparing *Wapl*^{+/+} pre-B cells to *Wapl*^{+/+} pro-B cells, a comparison referred to as having strong differences. I understand this is partially due

to loop callers and their issues in reproducibly reporting the same pixel as loops. **C)** At this point, all I can suggest is the authors clearly indicate where their conclusions are based on visual inspection instead of systematic comparisons or objective interpretation of the loop numbers they present as "our visual inspection of heatmaps suggested that this etc".

A) In the previous point-by-point reply, we showed published Hi-C contact matrices of chromosome 12 for the Pax5^{Δ/Δ} progenitors, wild-type (WT) pro-B cells, WT pre-B and WT mature B cells (Figure 5c; Hill et al., Nature 2020) as Figure 1 (for reviewer) to indicate by visual inspection that the chromosomal architecture is fundamentally different in WT pro-B cells compared with Pax5^{Δ/Δ} progenitors, WT pre-B and WT mature B cells. We have now statistically analyzed the relatedness of the Hi-C patterns of WT pro-B, WT pre-B and WT mature B cells by determining the Hi-C contact correlations between these cell types (Figure 2 for reviewer) by using the HiCRep.py method (Li et al., 2021, Bioinformatics). This new analysis clearly demonstrated that the WT pre-B and WT mature B cells are more closely related in their Hi-C structure compared with the WT pro-B cells, which are clearly the outliers.

Figure 2 (for reviewer). Hi-C contact correlations between the indicated B cell types were determined by the HiCRep.py method described in the Methods section. The correlation coefficient for each B cell comparison is shown.

B) The reviewer is right that the common loops between all three B cell comparisons (Extended Data Figures 3b,g and 4d) range from 47% to 58% of all loops called with the HiCCUPS algorithm. In this context, it is, however, important to point out that it has been published that stringent loop calling between replicate Hi-C experiments of the same cell line with HiCCUPS results in 63%-68% of common loop as shown in Figure 2k of Ardakany et al. (2020, Genome Biology 21, 256). Moreover, it has also been shown that relaxing the stringency of loop calling can result in up to 90% of common loops by comparing replicate Hi-C experiments (Figure 2A, Liu et al., 2021, Hereditas 158, 43). These published data clearly indicate that we should not overinterpret the percentage of common loop due to the inherent difficulties of currently available loop caller algorithms. In other words, loop calling is not a robust way of comparing Hi-C data. We also would like to point out that we used the term "unequivocally" only in the point-by-point reply of the first revision (23 December 23), but never throughout the entire manuscript.

C) The main conclusions of the revised manuscript are not only based on Hi-C contact matrices, but also on statistical analyses of their respective Hi-C data (Extended Data Figures 2 and 4) that we performed for the first revision of the manuscript. The conclusion of the data shown in Figure 1 (for reviewer, first point-by-point reply) were initially based on visual inspection. As these data have already been published, they will not be present in the final manuscript. To go beyond visual inspection of Figure 1 (for reviewer), we now demonstrate by Hi-C contact correlation analysis that the WT pro-B cells are distantly related in their Hi-C structure compared with WT pre-B and WT mature B cells (Figure 2; for reviewer). As requested by the reviewer, we have added "by visual inspection" in the manuscript, where appropriate.

REVIEWERS' COMMENTS

Reviewer #2 (expert in expert in 3D genome conformation, hi-C, micro-C):

I thank the authors for fully addressing the remaining concerns about the comparative analyses. If I may, I would suggest they remove the confusing sentence "Slight differences in the numbers of "common" loops within a cell pair resulted from very large loops lying close to each other." from the Methods. The minor differences are likely due to the size differences of loop anchors as HiCCUPS calls loops at multiple resolutions. No other comments or concerns.

RESPONSE TO REVIEWERS' COMMENTS

Enclosed, we submit our response to the comment of reviewer #2 (in black). Our reply is highlighted in blue.

Reviewer #2 (expert in expert in 3D genome conformation, hi-C, micro-C):

I thank the authors for fully addressing the remaining concerns about the comparative analyses. If I may, I would suggest they remove the confusing sentence "Slight differences in the numbers of "common" loops within a cell pair resulted from very large loops lying close to each other." from the Methods. The minor differences are likely due to the size differences of loop anchors as HiCCUPS calls loops at multiple resolutions. No other comments or concerns.

We thank the reviewer #2 for acknowledging that we did a good job in the 2nd revision and for favoring publication of our manuscript. As suggested by the reviewer, we have removed the confusing sentence at the end of the first paragraph on page 30.